# Adipocyte lysoplasmalogenase TMEM86A regulates plasmalogen homeostasis and protein kinase A-dependent energy metabolism

Yoon Keun Cho [1,7], Young Cheol Yoon [1,7], Hyeonyeong Im [1,7], Yeonho Son[1], Minsu Kim[1], Abhirup Saha[1], Cheoljun Choi [1], Jaewon Lee[1], Sumin Lee[1], Jae Hyun Kim[2], Yun Pyo Kang[1], Young-Suk Jung[3], Hong Koo Ha [4], Je Kyung Seong [5,8✉], James G. Granneman [6,8✉], Sung Won Kwon [1,8✉] & Yun-Hee Lee [1,8✉]

Dysregulation of adipose tissue plasmalogen metabolism is associated with obesity-related metabolic diseases. We report that feeding mice a high-fat diet reduces adipose tissue lysoplasmalogen levels and increases transmembrane protein 86 A (TMEM86A), a putative lysoplasmalogenase. Untargeted lipidomic analysis demonstrates that adipocyte-specific TMEM86A-knockout (AKO) increases lysoplasmalogen content in adipose tissue, including plasmenyl lysophosphatidylethanolamine 18:0 (LPE P-18:0). Surprisingly, TMEM86A AKO increases protein kinase A signalling pathways owing to inhibition of phosphodiesterase 3B and elevation of cyclic adenosine monophosphate. TMEM86A AKO upregulates mitochondrial oxidative metabolism, elevates energy expenditure, and protects mice from metabolic dysfunction induced by high-fat feeding. Importantly, the effects of TMEM86A AKO are largely reproduced in vitro and in vivo by LPE P-18:0 supplementation. LPE P-18:0 levels are significantly lower in adipose tissue of human patients with obesity, suggesting that TMEM86A inhibition or lysoplasmalogen supplementation might be therapeutic approaches for preventing or treating obesity-related metabolic diseases.

[1] College of Pharmacy and Research Institute of Pharmaceutical Sciences, Seoul National University, Seoul, Republic of Korea. [2] College of Pharmacy, Kangwon National University, Chuncheon, Gangwon-do, Republic of Korea. [3] Department of Pharmacy, College of Pharmacy, Research Institute for Drug Development, Pusan National University, Busan, Republic of Korea. [4] Department of Urology, Pusan National University Hospital, College of Medicine, Pusan National University, Busan, Republic of Korea. [5] Korea Mouse Phenotyping Center (KMPC), and Laboratory of Developmental Biology and Genomics, College of Veterinary Medicine, Seoul National University, Seoul, Republic of Korea. [6] Center for Molecular Medicine and Genetics, Wayne State University, Detroit, MI, USA. [7] These authors contributed equally: Yoon Keun Cho, Young Cheol Yoon, Hyeonyeong Im. [8] These authors jointly supervised this work: Je Kyung Seong, James G. Granneman, Sung Won Kwon, and Yun-Hee Lee. ✉email: snumouse@snu.ac.kr; jgranne@med.wayne.edu; swkwon@snu.ac.kr; yunhee.lee@snu.ac.kr

Adipose tissue is a crucial metabolic organ whose integrated metabolic, endocrine and immune functions are critical for systemic metabolic health[1]. Adipocytes have a complex, depot-specific lipidome that undergoes dynamic remodeling in response to systemic energy demands and nutritional status[2,3]. Among various lipid species, plasmalogens are a major component of adipocyte membrane phospholipids that are involved in the generation of lipid mediators[3,4]. Importantly, dysregulation of plasmalogen metabolism in adipocytes is associated with obesity-related metabolic disease[5,6] and previous studies have suggested regulatory roles of plasmalogen in mitochondrial oxidative metabolism in adipose tissue. For instance, adipose-specific disruption of peroxisomal biogenesis reduces plasmalogen biosynthesis and impairs mitochondrial oxidative metabolic activity. The effects are reversed by dietary supplementation of plasmalogens[7]. Nonetheless, the specific enzymatic pathways that maintain plasmalogen homeostasis in adipocytes are largely unknown.

The biosynthesis of plasmalogens begins in the luminal side of the peroxisomal membrane and intermediate products are further modified into plasmalogens in the endoplasmic reticulum (ER)[4]. Plasmalogens can be hydrolyzed by phospholipase A2 (PLA2), releasing fatty acids at the sn-2 position and lysoplasmalogens. The lysoplasmalogen can be further catabolized by phospholipase C (PLC), phospholipase D (PLD), and lysoplasmalogenase[4,6]. Moreover, lysoplasmalogens may be reacylated at sn-2 by transacylation, thus reforming plasmalogens[8]. Lysoplasmalogenases may be key enzymes that regulate lysoplasmalogen catabolism and lysoplasmalogen and plasmalogen homeostasis[6]. Lysoplasmalogenase was identified and characterized in rat liver microsomes as the enzyme that catalyzes the hydrolysis of the vinyl ether bond of choline[9] or ethanolamine[10] lysoplasmalogens, forming fatty aldehyde and glycerophosphocholine or glycerophosphoethanolamine, respectively. The enzyme was later found in rat brain microsomes[11] and in small intestinal epithelial cell microsomes[12]. The liver enzyme was purified[13] and the gene was identified as TMEM86B[6]. TMEM86A is a close homolog of TMEM86B;[6] however, its potential lysoplasmalogenase activity and function in adipose tissue have not been investigated.

The beta-adrenergic/cyclic adenosine monophosphate (cAMP)/protein kinase A (PKA) signalling pathway plays a crucial role in controlling the catabolic phenotype of adipose tissues by regulating lipolysis[14], oxidative gene transcription[15], and mitochondrial oxidative metabolism[16]. Indeed, activation of the pathway at the level of receptor[17,18], cAMP levels[19], or PKA[20] in adipocytes expands oxidative metabolism, increases energy expenditure, and protects against diet-induced obesity[16]. Previously, Williams and Ford investigated the direct activation of purified PKA holoenzyme from heart tissue, and found that lysoplasmenylcholine activated the purified complex[21]. However, whether and how lysoplasmalogens might influence PKA signalling in adipocytes has not been investigated.

In the current study, we show that high-fat feeding increased TMEM86A protein levels in adipocytes, suggesting that TMEM86A regulates levels of plasmalogen-derived lipid mediators during adipocyte hypertrophy. We provide evidence that TMEM86A is a bona fide lysoplasmalogenase by comprehensive global phospholipid profiling and direct enzymatic assays in gain- and loss-of-function analyses. We also addressed the physiological roles of adipocyte TMEM86A and its underlying molecular mechanisms in vitro and in adipocyte-specific TMEM86A KO (TMEM86A AKO) mice. We discovered that genetic inactivation of adipocyte TMEM86A increases oxidative metabolism in adipose tissues and improves systemic metabolism during high-fat feeding. Mechanistically, loss of TMEM86A potentiates PKA signalling by inhibiting phosphodiesterase 3B (PDE3B), the major

enzyme that degrades cAMP in adipocytes. Importantly, treating mice with lysoplasmalogen, the substrate degraded by TMEM86A, protected mice from high-fat diet (HFD)-induced metabolic dysfunction. Collectively, these results indicate that lysoplasmalogens are key regulators of catabolic signalling in adipocytes, and suggest that TMEM86A might be targeted for obesity-related metabolic disease.

## Results

**High-fat diet induces TMEM86A expression in white adipose tissue.** We performed RNA-seq analysis of gonadal white adipose tissue (gWAT) of mice fed a normal chow diet (NCD) or HFD for 8 weeks and analyzed the expression levels of genes involved in plasmalogen metabolism (Fig. 1a). Among these genes, the expression of *Tmem86a* was significantly upregulated in gWAT of HFD-fed mice compared to that in mice fed a NCD (Fig. 1a), suggesting the involvement of TMEM86A in the pathologic response of the adipose tissue induced by overnutrition. RT-qPCR and immunoblot analyses showed that HFD feeding increased TMEM86A, but not TMEM86B, expression in WAT (Fig. 1b–g, and Supplementary Fig. 1) consistent with the RNA-seq data. Furthermore, publicly available transcriptome-profiling (GEO: GSE94753) indicates that *TMEM86A* expression is upregulated in abdominal subcutaneous WAT from female patients with obesity manifesting insulin resistance (OIR) compared to individuals without obesity (NO) (Fig. 1h).

Adipose tissue is composed of numerous cell types. To determine the cellular distribution of TMEM86A, we fractionated adipose tissue using a combination of flotation (adipocytes) and magnetic bead separation (F4/80+ and PDGFRα + cells) (Supplementary Fig. 2a, b). We found that *Tmem86a* mRNA was heavily enriched in adipocytes in both iWAT and gWAT (Fig. 1i, j).

**TMEM86A functions as lysoplasmalogenase in adipocytes.** TMEM86A and TMEM86B are members of the YhhN family proteins found in various species including bacteria[22]. Although lysoplasmalogenase activity of TMEM86A is predicted by its sequence similarity with TMEM86B[22] (Supplementary Fig. 3), the impact of TMEM86A on lysoplasmalogen metabolism has not been evaluated. To do so, we performed global unbiased phospholipid profiling of TMEM86A overexpressing C3H10T1/2 adipocytes and controls (Fig. 2a). In this analysis, we used adipocytes differentiated from C3H10T1/2 cells, a cloned murine embryo fibroblast cell line[23]. This untargeted lipidomics analysis included the following classes of phospholipids: lysophosphatidylcholine (LPC), lysophosphatidylethanolamine (LPE), phosphatidylcholine (PC), phosphatidylethanolamine (PE), plasmanyl LPC (annotated as LPC O-), plasmanyl PC (annotated as PC O-), plasmenyl LPC (annotated as LPC P-), plasmenyl LPE (annotated as LPE P-), plasmenyl PC (annotated as PC P-), and plasmenyl PE (annotated as PE P-) (Supplementary Data 1). Acid hydrolysis method was utilized to distinguish between plasmanyl PC and plasmenyl PC.

In total, untargeted phospholipid profiling detected 227 phospholipids including 8 species of lysoplasmalogen and 32 species of plasmalogen with high confidence (Supplementary Data 1). Adipocytes overexpressing TMEM86A were clearly separated from mock controls in the principal component analysis (PCA) plot (Fig. 2b). The downregulated lipid species in TMEM86A overexpressing cells (|fold change| > 2, FDR < 0.05) were lysoplasmalogens or plasmalogens, indicated by volcano plot, heatmap (Fig. 2c, d and Supplementary Fig. 4), and the normalized intensity (Fig. 2e, f and Supplementary Fig. 5a, b). TMEM86A overexpression upregulated plasmanyl PC and

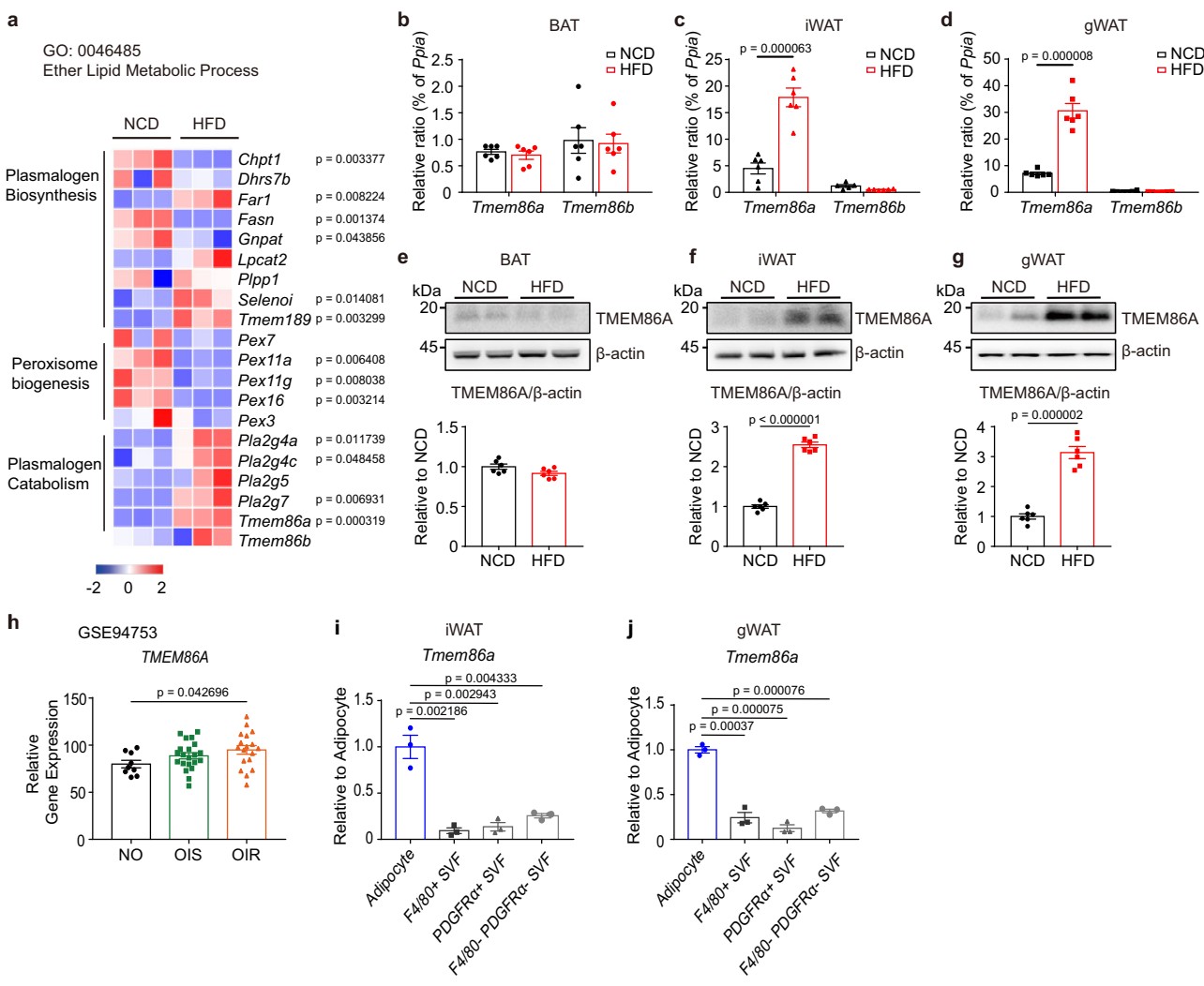

**Fig. 1 TMEM86A is abundantly expressed in WAT and upregulated by HFD feeding. a** Heatmap of RNA-seq data showing the effects of 8 weeks of high-fat diet (HFD) feeding on genes involved in ether lipid metabolism (GO: 0046485) in gWAT. (NCD: normal chow diet, *| fold change (FC) | > 2; $p < 0.05$) **b–d** qPCR analysis of *Tmem86a* and *Tmem86b* in BAT, iWAT and gWAT of mice fed a NCD or HFD for 8 weeks. $n = 6$. **e–g** Immunoblot analysis of TMEM86A expression. $n = 6$. **h** Analysis of *TMEM86A* expression in human adipose tissue (non-obesity (NO): $n = 9$; insulin-sensitive obesity (OIS): $n = 21$; insulin-resistant obesity (OIR): $n = 18$), determined by publicly available transcriptomic data (Gene Expression Omnibus (GEO) repository, accession number GSE94753). qPCR analysis of *Tmem86a* expression in adipocytes and stromal vascular fraction (SVF; F4/80+, PDGFRα+, or F4/80-PDGFRα-) isolated from iWAT (**i**) and gWAT (**j**) of mice. $n = 3$. Each point represents a biological replicate. Data are presented as the mean ± SEM. Statistical significance was determined using the unpaired, two-tailed *t*-test in **a–j**. Source data are provided as a Source Data file.

plasmanyl PE levels, while no significant changes were observed for LPC, LPE, PC, and plasmanyl LPC (Fig. 2d).

As an independent approach, TMEM86A overexpressing C3H10T1/2 adipocytes were challenged with LPE P-18:0, and LPE P-18:0 levels were measured by LC-MS analysis (Fig. 2g). TMEM86A overexpression significantly increased LPE P-18:0 catabolism calculated by subtracting residual LPE P-18:0 from the initial levels in the conditioned media (Fig. 2g).

The structure of TMEM86A has not been determined; however, a high confidence AlphaFold computational model[24] predicts a protein with 8 transmembrane regions (Fig. 2h). Consistent with this prediction and experiments with TMEM86B[6], we found that green fluorescent protein (GFP)-tagged TMEM86A strongly colocalized with ER Tracker staining in C3H10T1/2 cells both before and after differentiation (Fig. 2i and Supplementary Fig. 6a, b). Evolutionary analysis of the YhhN family proteins by Jurkowitz, et al.[22]. identified several residues, including D82 and D190, that are absolutely conserved between bacterial and mammalian YhhN lysoplasmalogenases

(Supplementary Fig. 3) and the AlphaFold model juxtaposed potential catalytic histidine and aspartate residues within the predicted transmembrane region (Fig. 2h). We found that D82A or D190A mutation of TMEM86A reduced lysoplasmalogenase activity in HEK293T cells (Fig. 2j, k), further implicating these residues in catalytic activity.

**Adipocyte-specific TMEM86A KO increases lysoplasmalogen content.** To investigate the physiological roles of TMEM86A in adipose tissue, we established adipocyte-specific TMEM86A KO model (TMEM86A AKO) by crossing *Tmem86a*flox/flox (TMEM86A fl/fl) mice with adipoq-Cre mice (Fig. 3a and Supplementary Fig. 7). TMEM86A KO in BAT, iWAT, and gWAT was confirmed by western blot analysis (Fig. 3b–e).

We then performed untargeted lipidomics analysis of BAT, iWAT, and gWAT from wild type (WT) and TMEM86A AKO mice to determine the effects of TMEM86A AKO on phospholipid composition, including plasmalogen metabolism. Detection

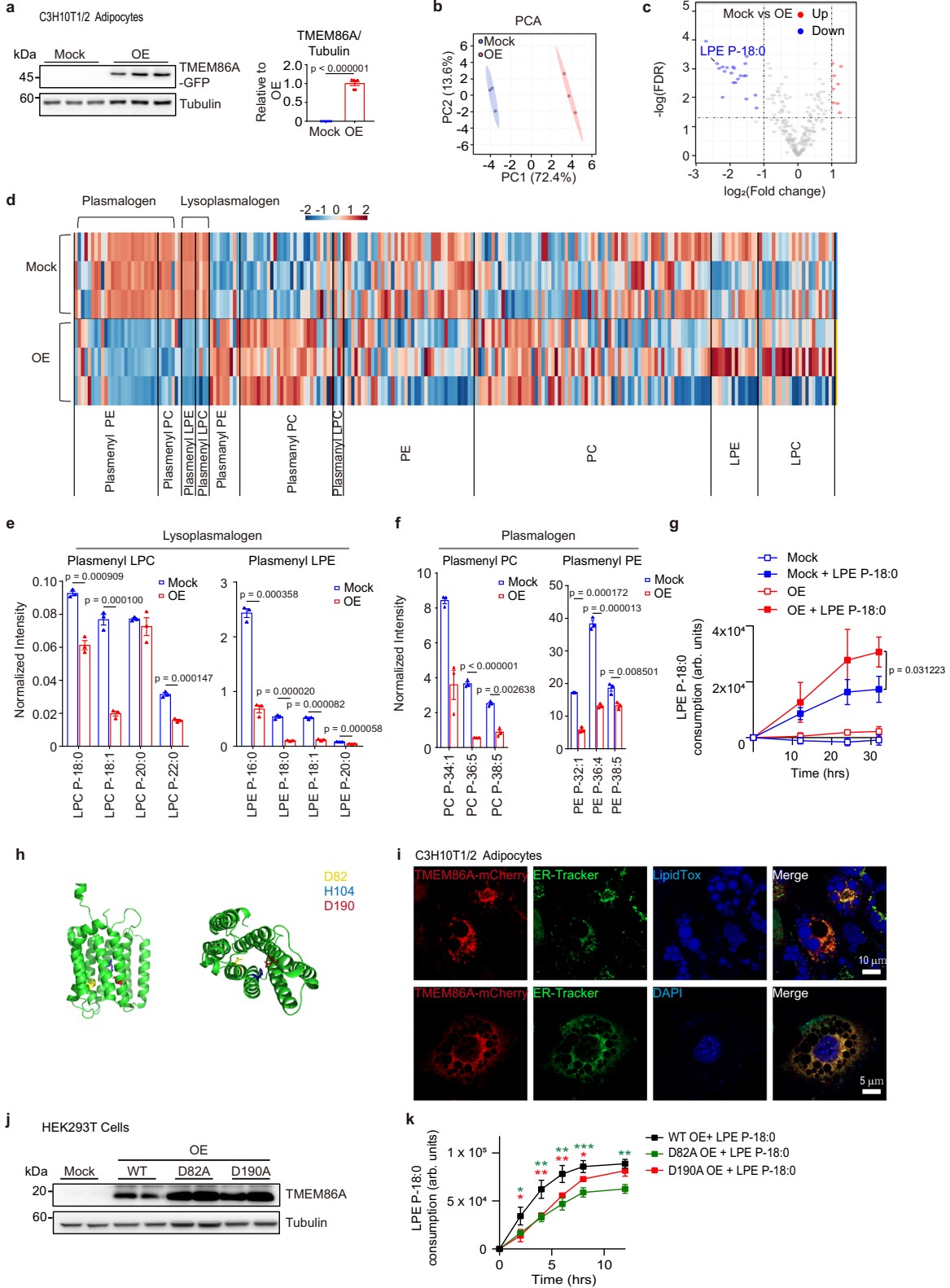

of lysoplasmalogens in adipose tissue samples can be challenging due to the abundant neutral lipids which lead to ion suppression of phospholipids[25]. Therefore, we used phospholipid-enriched fractions obtained from the three-phase liquid extraction method[25] to remove the excess amount of neutral lipids (e.g. triglyceride) in adipocytes. In total, 243 phospholipids including 10 lysoplasmalogens (plasmenyl LPC and plasmenyl LPE) and 42

plasmalogens (plasmenyl PC and plasmenyl PE) were identified with high confidence (Supplementary Data 2).

PCA revealed clear differences in lipid profiles between genotypes and among adipose tissue depots (Fig. 3f and Supplementary Fig. 8a). Principal component (PC) 1 separated BAT and WAT, while PC 2 separated iWAT and gWAT (Supplementary Fig. 8b). Each PCA plot of all three adipose tissue

**Fig. 2 TMEM86A regulates adipocyte lysoplasmalogen metabolism. a** Immunoblot analysis of TMEM86A expression in C3H10T1/2 adipocytes overexpressing TMEM86A (OE) or controls (mock) $n = 3$. Data are presented as the mean ± SEM. **b–d** PCA, volcano plot, and heatmap of phospholipid profiling of TMEM86A-overexpressing C3H10T1/2 adipocytes (OE) or controls (mock). **e, f** Normalized intensity of lysoplasmalogens and plasmalogens from phospholipid profiling of TMEM86A-overexpressing C3H10T1/2 adipocytes or controls. $n = 3$. (Intensity was normalized by median, *FDR < 0.05). Data are presented as the mean ± SEM. **g** LPE P-18:0 consumption levels of TMEM86A-overexpressing C3H10T1/2 adipocytes or controls. Cells were treated with LPE P-18:0 (1 μM) and then LPE P-18:0 in conditioned media were measured by LC-MS at indicated time. LPE P-18:0 consumption was calculated by subtracting LPE P-18:0 peak area at each time point from the initial value followed by normalization with a total protein quantity of cells. $n = 3$. Data are presented as the mean ± SD. **h** AlphaFold 3D model of human TMEM86A protein (AF-Q5BLD1-F1-model_v2.pdb). **i** Representative images of C3H10T1/2 adipocytes overexpressing GFP-tagged TMEM86A, stained with ER-Tracker. Nuclei and lipid were counterstained with DAPI and LipidTox, respectively. Scale bars = 10 μm. $n = 3$. **j** Immunoblot analysis of TMEM86A expression in HEK293T cells overexpressing WT, D82A mutant, or D190A mutant TMEM86A. $n = 3$. **k** LPE P-18:0 consumption levels of HEK293T cells overexpressing WT and mutant TMEM86A. $n = 3$. (WT OE + LPE P-18:0 vs. D82A OE + LPE P-18:0: $p = 0.036525$, $p = 0.002498$, $p = 0.001757$, $p = 0.000735$, or $p = 0.001335$, respectively for 2, 4, 6, 8, or 12 h; WT OE + LPE P-18:0 vs. D190A OE + LPE P-18:0: $p = 0.019168$, $p = 0.003337$, $p = 0.00967$, $p = 0.02689$, or $p = 0.251998$, respectively for 2, 4, 6, 8, or 12 h). $p$ value was calculated by comparing each mutant group with WT OE + LPE P-18:0 as a control group. Data are presented as the mean ± SD. Each point represents a biological replicate. Statistical significance was determined using the unpaired, two-tailed $t$-test in **a**, **e**, **f**, **g** and using ANOVA (one-way, Bonferroni's test) in **k**. Source data are provided as a Source Data file.

depots represented well separation of WT and TMEM86A AKO groups with little overlap (Supplementary Fig. 8c). Additionally, PC 3 clearly distinguished the lipidomic signatures of TMEM86A AKO vs. WT mice (Fig. 3f). Furthermore, heatmap analysis revealed that the overall levels of lysoplasmalogens were significantly higher in all three depots of adipose tissue from TMEM86A AKO mice compared to the levels in WT mice (Fig. 3g and Supplementary Fig. 9). Additionally, we observed higher LPE and PE levels, but lower levels of plasmanyl PC, plasmanyl LPC, and plasmalogens in BAT than in WAT (Fig. 3g and Supplementary Fig. 9). This result is consistent with a previous report[2] showing higher PE and lower plasmanyl PC and plasmenyl PE in brown versus white adipose tissue.

Next, we examined phospholipid species that were differentially regulated in distinct depots (BAT, iWAT, and gWAT) of TMEM86A AKO mice. Volcano plots demonstrated that significantly upregulated phospholipids (|fold change| > 2, FDR < 0.05) were lysoplasmalogens or plasmalogens (Fig. 3h–j and Supplementary Fig. 10a–c). Moreover, our results revealed that TMEM86A AKO elevated the levels of most lysoplasmalogens and plasmalogens in adipose tissues regardless of the anatomical location (Fig. 3k–p and Supplementary Fig. 11a–f). Among the differentially regulated lipid species, only 1-(1Z-octadecenyl)-sn-glycero-3-phosphocholine (LPC P-18:0) and 1-(1Z-octadecenyl)-sn-glycero-3-phosphoethanolamine (LPE P-18:0) showed a |fold change| > 2 and FDR < 0.05 in all adipose tissue types (Fig. 3k–p). In line with various analyses, LPC P-18:0 and LPE P-18:0 exhibited the highest or second highest normalized levels compared to other lipid species in their respective lipid subclasses (Fig. 3k–p). These KO mouse studies provide strong evidence that TMEM86A hydrolyzes lysoplasmalogens, thus controlling levels of the lysolipid.

**Adipocyte-specific TMEM86A KO upregulates PKA downstream signalling.** Plasmalogen levels are positively correlated with mitochondrial oxidative activity[7,26]. To determine whether lipid remodeling by TMEM86A AKO is associated with mitochondrial oxidative metabolism, we analyzed mitochondrial content and activity in WT and TMEM86A AKO mice. Immunoblot analysis demonstrated that TMEM86A AKO increases levels of mitochondrial proteins involved in oxidative phosphorylation and thermogenesis, including medium-chain acyl-CoA dehydrogenase (MCAD), ATP synthase complex 5 (ATP5A), ubiquinol-cytochrome-c reductase complex core protein 2 (UQCRC2), succinate dehydrogenase iron-sulfur subunit (SDHB), NADH dehydrogenase ubiquinone 1 beta subcomplex subunit 8 (NDUFB8), cytochrome c oxidase subunit 4 (COXIV), and uncoupling protein 1 (UCP1) in BAT and WAT (Fig. 4a–d).

Previous studies have reported that lysoplasmalogen facilitates PKA downstream signalling;[21] thus, we hypothesized that lysoplasmalogen-mediated activation of PKA signalling might contribute to enhanced mitochondrial activity in the adipose tissue of TMEM86A AKO mice. In line with previous findings, we found that TMEM86A AKO significantly increased phosphorylation of HSL and CREB in BAT and WAT (Fig. 4a–d).

Measurement of whole-body energy metabolism by indirect calorimetry analysis indicated that TMEM86A AKO increases $VO_2$, $VCO_2$, and energy expenditure (Fig. 4e–h). Food intake and activity were not affected by TMEM86A AKO (Fig. 4i, j, Supplementary Fig. 12c). TMEM86A AKO mice exhibited significantly lower body weight (Supplementary Fig. 12a) percentage body fat, and increased percentage lean mass (Fig. 4k). TMEM86A AKO exhibited increased surface temperature when exposed to cold (4 °C) conditions (Fig. 4l), and in situ staining with triphenyltetrazolium chloride (TTC) demonstrated increased mitochondrial electron transport in all adipose tissue depots (Fig. 4m, n). Furthermore, we found that TMEM86A AKO upregulated the expression of several thermogenic genes including *Ppargc1a*, *Ucp1*, *Cox8b* and *Dio2* that are largely regulated by PKA activation (Supplementary Fig. 13a–c). These results are consistent with electron micrographs demonstrating that TMEM86A AKO increases number of mitochondria in adipocytes of BAT, iWAT, and gWAT (Fig. 4o–q). Collectively, these results showed that deletion of TMEM86A in adipocytes leads to a marked improvement in energy expenditure and mitochondrial activity that involves upregulation of PKA signalling.

**In vitro TMEM86A KO increases PKA downstream signalling in adipocytes.** Next, we examined the cell-autonomous effects of TMEM86A KO using primary adipocytes differentiated from pre-adipocytes isolated from the adipose tissues of WT or TMEM86A AKO mice. Consistent with the in vivo results, immunoblot analysis results showed that TMEM86A AKO increased the phosphorylation of HSL and CREB, as well as mitochondrial protein levels (MCAD, ATP5A, UQCRC2, SDHB, NDUFB8, and COXIV) (Fig. 5a–d). Interestingly, the effects of TMEM86A AKO on mitochondrial protein expression were most robust in primary white adipocytes derived from gWAT (Fig. 5a–d) and these effects correlated with increased mitochondrial respiration (Fig. 5e, f).

As a complementary approach, we overexpressed TMEM86A in C3H10T1/2 adipocytes as detailed above. We found that reduced phosphorylation of HSL, CREB, and PKA substrates, and

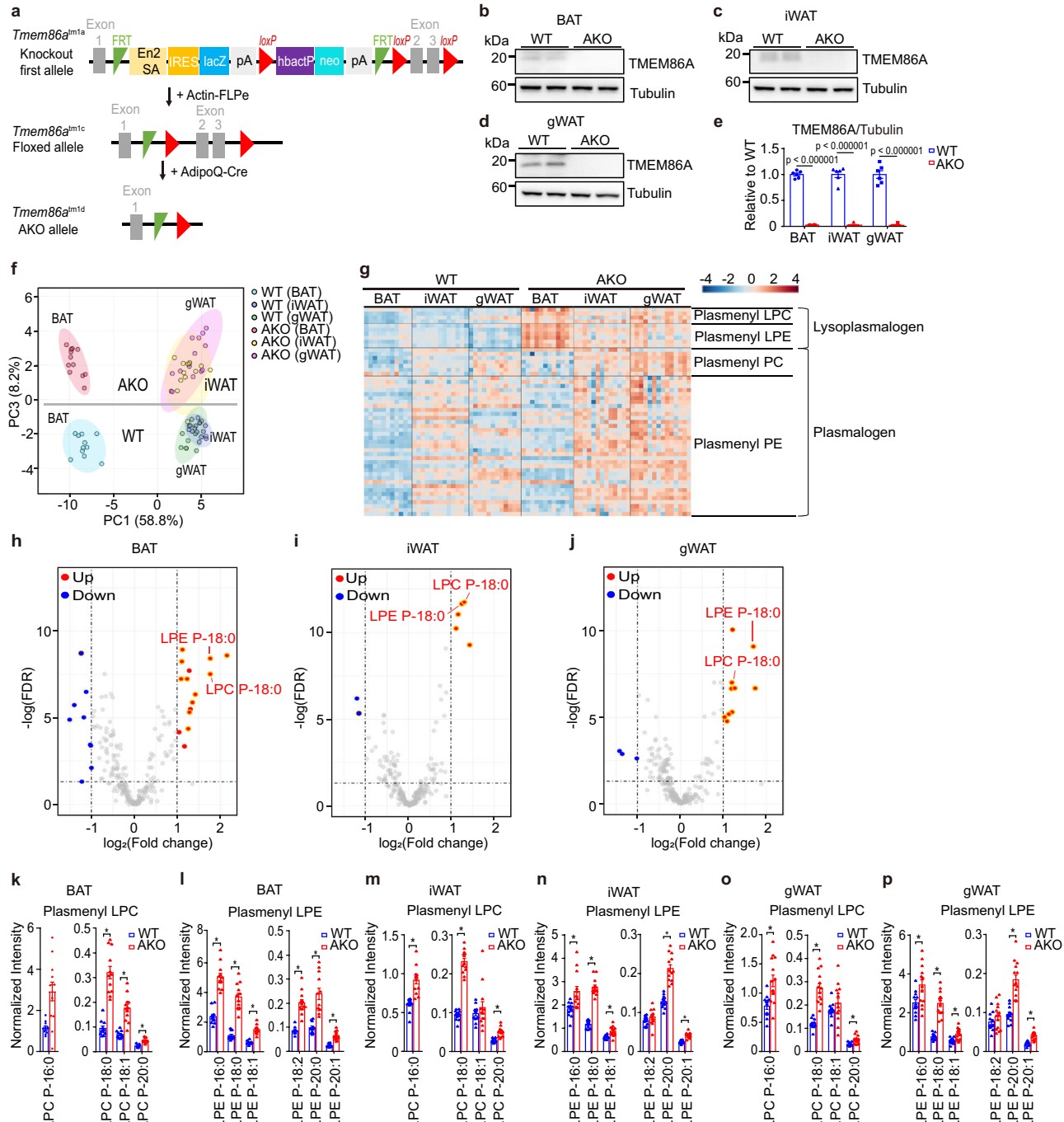

**Fig. 3 Untargeted phospholipid profiling of adipose tissue of adipocyte-specific TMEM86A KO mice. a** Schematic diagram showing generation of adipocyte-specific TMEM86A knockout mice (TMEM86A AKO). **b–e** Immunoblot analysis of TMEM86A expression in adipose tissue of WT and TMEM86A AKO mice. $n = 6$. **f** PCA plots showing a separation of clusters of WT and TMEM86A AKO samples. Heatmap analysis (**g**) and volcano plots (**h–j**) of untargeted phospholipid profiling of BAT, iWAT and gWAT from WT and TMEM86A AKO mice. Lysoplasmalogens are highlighted in yellow in **h–j**. BAT WT: $n = 11$, KO: $n = 12$; iWAT WT: $n = 13$, KO: $n = 13$; gWAT WT: $n = 12$, KO: $n = 14$. **k–p** Normalized intensities of lysoplasmalogens in BAT, iWAT and gWAT of WT and TMEM86A AKO mice (Intensity was normalized by median, *FDR < 0.05). BAT WT: $n = 11$, KO: $n = 12$; iWAT WT: $n = 13$, KO: $n = 13$; gWAT WT: $n = 12$, KO: $n = 14$. (WT BAT vs. AKO BAT: $p < 0.000001$ for LPC P-18:0, LPC P-18:1, LPC P-20:0, LPE P-16:0, LPE P-18:0, LPE P-18:1, LPE P-18:2, LPE P-20:0, and LPE P-20:1; WT iWAT vs. AKO iWAT: $p = 0.000003$ for LPC P-16:0, $p = 0.001422$ for LPE P-16:0, $p = 0.000835$ for LPE P-18:1, $p < 0.000001$ for LPC P-18:0, LPC P-20:0, LPE P-18:0, LPE P-20:0, and LPE P-20:1; WT gWAT vs. AKO gWAT: $p = 0.001322$ for LPC P-16:0, $p = 0.000694$ for LPC P-20:0, $p = 0.012532$ for LPE P-16:0, $p = 0.000865$ for LPE P-18:1, $p < 0.000001$ for LPC P-18:0, LPE P-18:0, LPE P-20:0, and LPE P-20:1) Each point represents a biological replicate. Data are presented as the mean ± SEM. Statistical significance was determined using the unpaired, two-tailed $t$-test in **a** and **k–p**. Source data are provided as a Source Data file.

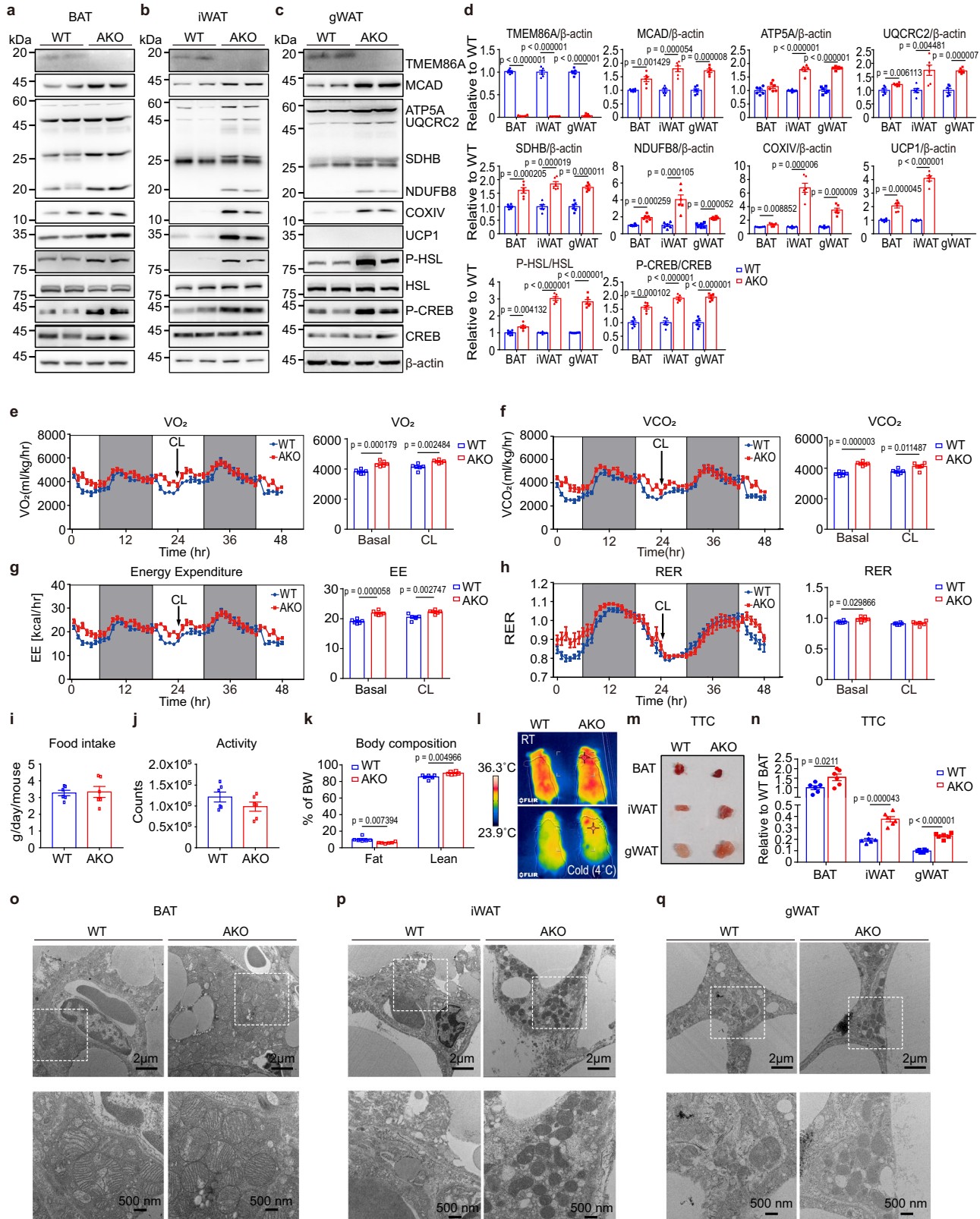

**Fig. 4 Adipocyte-specific knockout of TMEM86A enhances PKA-signalling pathway and mitochondrial oxidative metabolism in adipose tissue.**
**a–d** Immunoblot analysis of BAT, iWAT and gWAT of WT and TMEM86A AKO mice. $n = 6$. **e–h** Indirect calorimetry analysis. Arrows indicates CL316,243 (CL) injection. EE: energy expenditure; RER: respiratory exchange ratio; $VO_2$: rate of oxygen consumption; $VCO_2$: rate of carbon dioxide production. $n = 6$. **i**, **j** Monitoring of food intake and activity. **k** Body fat and lean mass. $n = 6$. **l** Infrared thermal images showing skin temperature at room temperature or after 60 min of cold exposure (4 °C). $n = 4$. **m**, **n** Triphenyltetrazolium chloride (TTC) staining of BAT, iWAT, and gWAT and quantification. $n = 6$. **o–q** Electron microscopy of BAT, iWAT and gWAT. Boxed regions are magnified. $n = 6$. Each point represents a biological replicate. Data are presented as the mean ± SEM. Statistical significance was determined using the unpaired, two-tailed $t$-test in **d–j** and **n**. Source data are provided as a Source Data file.

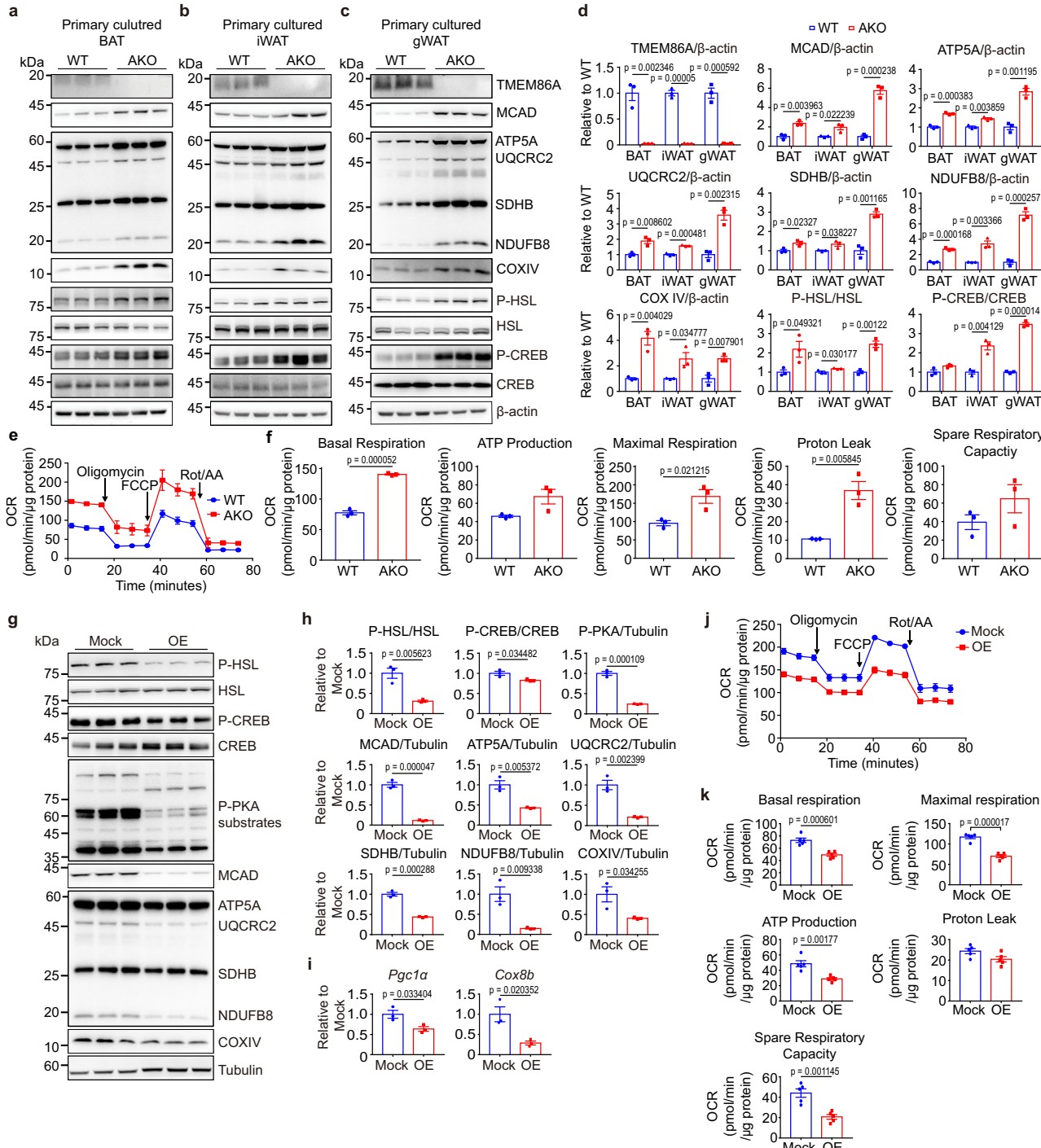

**Fig. 5 In vitro modulation of TMEM86A expression regulated PKA signalling pathway and mitochondrial oxidative metabolism of adipocytes.**
**a–d** Immunoblot analysis of adipocytes differentiated from precursors obtained from BAT, iWAT and gWAT of WT and TMEM86A AKO mice. $n = 3$.
**e, f** Oxygen consumption rate measurement of adipocytes differentiated from precursors obtained from gWAT of WT and TMEM86A AKO mice. $n = 3$.
**g–i** Immunoblot and qPCR analyses of TMEM86A overexpression effects in C3H10T1/2 adipocytes. $n = 3$. **j, k** Oxygen consumption rate of TMEM86A overexpressing C3H10T1/2 adipocytes or controls. $n = 5$. Each point represents a biological replicate. Data are presented as the mean ± SEM. Statistical significance was determined by the unpaired, two-tailed $t$-test in **d–f**, **h–i**, and **k**. Source data are provided as a Source Data file.

expression of PKA-target genes *Ppargc1a* and *Cox8b* that regulate mitochondrial biogenesis (Fig. 5g–i). As expected, TMEM86A overexpression lowered levels of mitochondrial proteins (MCAD, ATP5A, UQCRC2, SDHB, NDUFB8, and COXIV) (Fig. 5g, h) and reduced basal, maximal and ATP synthesis-dependent oxygen consumption as well as spare respiratory capacity (Fig. 5j, k).

**TMEM86A KO protects against HFD-induced metabolic dysfunction.** Next, we examined whether the deletion of TMEM86A protects mice from obesity-induced metabolic dysfunction. TMEM86A AKO reduced HFD-induced body weight gain over an 8-week feeding period (Fig. 6a). TMEM86A AKO greatly reduced weights of iWAT, gWAT and liver of HFD-fed mice

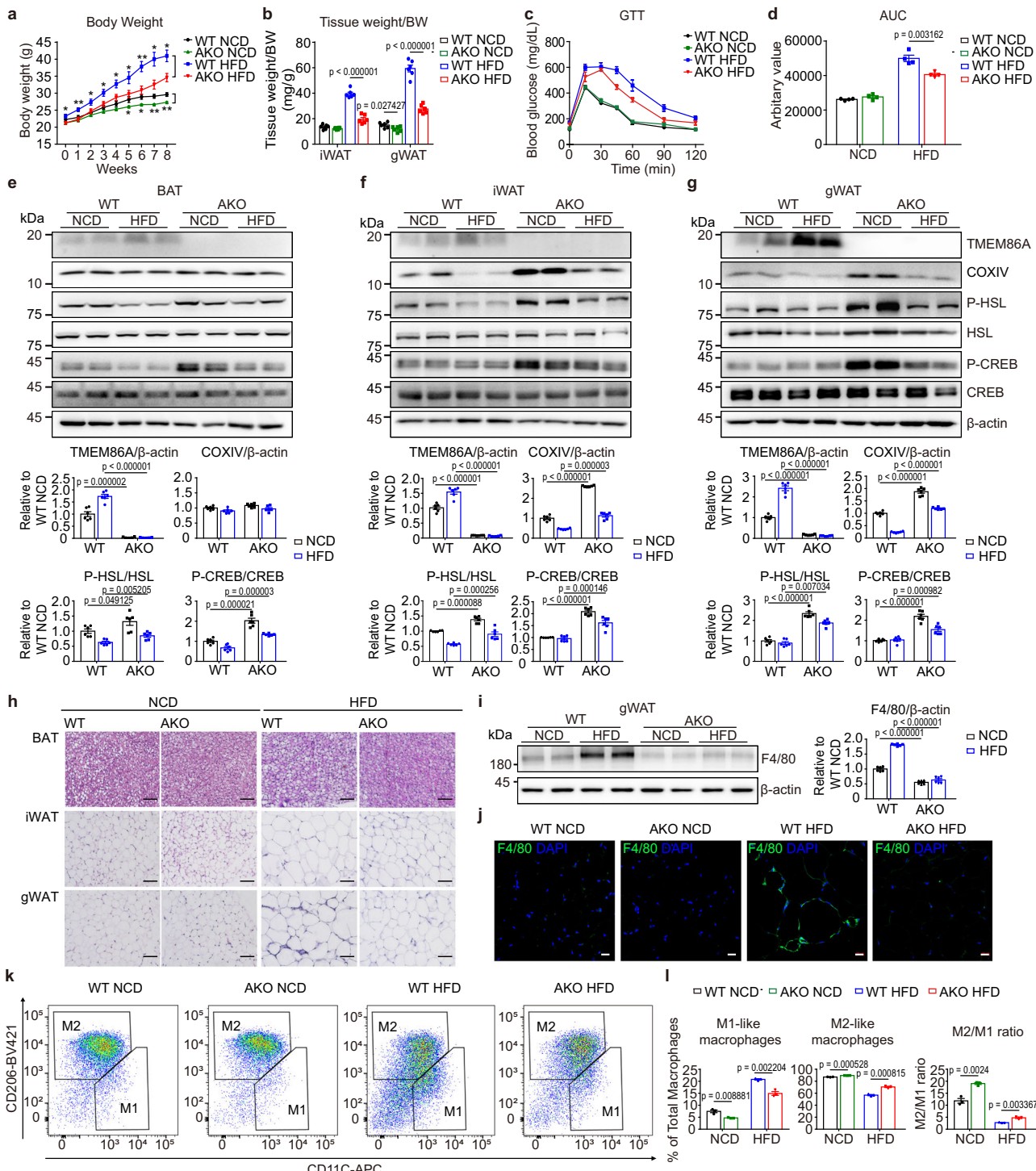

**Fig. 6 Adipocyte-specific TMEM86A KO protects mice against HFD-induced metabolic dysfunction and inflammation. a** Body weight monitoring of WT and TMEM86A AKO mice during 8-weeks of HFD feeding. $n = 6$. (WT HFD vs. KO HFD: $p = 0.015084$, $p = 0.006167$, $p = 0.042557$, $p = 0.026351$, $p = 0.019894$, $p = 0.040475$, $p = 0.004757$, $p = 0.010575$, or $p = 0.1215$, respectively for week 0, 1, 2, 3, 4, 5, 6, 7, or 8; WT NCD vs. KO NCD: $p = 0.039863$, $p = 0.010424$, $p = 0.003051$, or $p = 0.003318$, respectively for week 5, 6, 7, or 8). **b** Weights of adipose tissue of WT and TMEM86A AKO mice fed a NCD or HFD. $n = 6$. **c, d** Intraperitoneal glucose tolerance test. $n = 4$. **e–g** Immunoblot analysis of BAT, iWAT and gWAT. $n = 6$. **h** H&E staining of paraffin sections of BAT, iWAT and gWAT. $n = 6$. Scale bar = 50 μm. **i** Immunoblot analysis of F4/80 in gWAT. $n = 6$. **j** Immunofluorescence staining of F4/80 in paraffin sections of gWAT. $n = 6$. Scale bar = 20 μm. **k, l** Flow cytometric analysis of CD45+CD11b+CD64+CD11C+ cells (M1-like macrophages) and CD45+CD11b+CD64+CD206+ cells (M2-like macrophages) in gWAT. $n = 3$. Each point represents a biological replicate. Data are presented as mean ± SEM. Statistical significance was determined using the unpaired, two-tailed t-test in **a**, **b**, **d**, **e–g**, **i**, Source data are provided as a Source Data file.

compared to WT mice (Fig. 6b and Supplementary Fig. 14a), and improved glucose tolerance (Fig. 6c, d) and insulin sensitivity (Supplementary Fig. 14b, c). Phosphorylation levels of PKA signalling downstream targets (CREB and HSL) were modestly upregulated in TMEM86A AKO mice in both NCD and HFD groups (Fig. 6e–g). Histological observation showed smaller sizes of adipocytes and reduced lipid accumulation in liver of TMEM86A AKO groups compared to WT controls (Fig. 6h, Supplementary Fig. 14d, e). These results suggest that the anti-obesity effects of TMEM86A AKO are mediated in part by PKA signalling-mediated increase in mitochondrial activity.

Obesity is associated with the accumulation of pro-inflammatory macrophages in adipose tissue. We found that TMEM86A AKO prevented HFD-induced accumulation of the macrophage marker F4/80 in gWAT (Fig. 6i), as well as the appearance of F4/80+ crown-like structures (Fig. 6j). As expected, flow cytometric analysis of stromal vascular fraction (SVF) isolated from gWAT demonstrated that TMEM86A deletion prevented the diet-induced accumulation of total macrophages (CD45$^+$CD11b$^+$CD64$^+$ cells) (Supplementary Fig. 15a) and reduced the proportions of M1-like pro-inflammatory (CD45$^+$CD11b$^+$CD64$^+$CD11C$^+$ cells) relative to M2-like anti-inflammatory phenotypes (CD45$^+$CD11b$^+$CD64$^+$CD206$^+$ cells) (Fig. 6k, l).

**LPE P-18:0 treatment activates PKA signalling via PDE3B inhibition.** Next, we examined the effects of LPC P-18:0 and LPE P-18:0 that were identified by phospholipid profiling as major lysoplasmalogens increased by TMEM86A AKO and decreased by TMEM86A overexpression in adipocytes. Treatment of C3H10T1/2 adipocytes with LPC P-18:0 or LPE P-18:0 significantly increased levels of phosphorylated HSL, CREB, and PKA substrates (Fig. 7a, b). Co-treatment of LPC P-18:0 or LPE P-18:0 with H89, a PKA inhibitor, suppressed phosphorylation of HSL and CREB (Fig. 7c, d), confirming the involvement of PKA signalling in the lysoplasmalogen-mediated effects.

Consistent with PKA activation, treatment with either LPC P-18:0 or LPE P-18:0 for 24 h elevated intracellular cAMP levels in C3H10T1/2 adipocytes (Fig. 7e). TMEM86A overexpression, which reduces lysoplasmalogen levels, also lowered cAMP levels (Supplementary Fig. 16). LPE P-18:0 did not appear to enhance the coupling of beta adrenergic receptors to the stimulatory G-protein of adenylyl cyclase (GNAS), as assessed by mVenus-tagged mini Gs protein[27] assay in HEK 293 T cells (Supplementary Fig. 17). Next, we tested whether LPE P-18:0 inhibits phosphodiesterase (PDE) activity, which is the major mechanism for degrading cAMP in adipocytes. PDE3B activity was tested as our RNAseq analysis (GSE 182930) indicated that Pde3b was the most abundantly expressed isoform in gWAT (Supplementary Fig. 18). We found that LPE P-18:0 reduced PDE3B activity to the levels comparable to the effects of a non-specific PDE inhibitor, isobutylmethylxanthine (IBMX) (Fig. 7f). In addition, co-treatment of LPE P-18:0 and cilostamide, a selective PDE3B inhibitor, did not show any additive effects on phosphorylation of HSL and CREB, suggesting that the effects of LPE P-18:0 require PDE3 activity (Supplementary Fig. 19a, b). Collectively, these data support the hypothesis that LPE P-18:0 can facilitate PKA-dependent signalling pathways by inhibiting PDE3B activity.

Consistent with the results above, longer-term treatment (48 h) of C3H10T1/2 adipocytes with LPC P-18:0 or LPE P-18:0 upregulated expression of mitochondrial enzymes, including MCAD, ATP5A, UQCRC2, SDHB, NDUFB8, and COXIV (Fig. 7g, h). In addition, adipocytes treated with LPE P-18:0 exhibited elevated maximal, basal, proton leak-related, and ATP production-related oxygen consumption rates as well as spare respiratory capacity (Fig. 7i, j).

**LPE P-18:0 treatment protects against diet-induced obesity.** Lastly, we examined the in vivo effects of LPE P-18:0 treatment. As expected, HFD feeding significantly reduced LPE P-18:0 levels in iWAT and gWAT (Supplementary Fig. 20). LC/MS/MS analysis confirmed that two weeks of LPE P-18:0 treatment increased LPE P-18:0 levels in serum, to a level comparable to TMEM86A AKO mice (Fig. 8a). Glucose intolerance and insulin resistance caused by HFD were improved by LPE P-18:0 treatment (Fig. 8b, c and Supplementary Fig. 21d). LPE P-18:0 treatment reduced body weight and fat mass, and increased VO$_2$, and energy expenditure, without effects on food intake and activity (Supplementary Fig. 21a–c). In line with in vitro data, immunoblot analysis demonstrated that LPE P-18:0 upregulated PKA signalling and increased mitochondrial protein levels in adipose tissue (Fig. 8d–g).

Untargeted phospholipid profiling was performed using human subcutaneous adipose tissue (SC) ($n = 48$). To analyze the change in lipid species levels in accordance with BMI, 48 individuals were divided into four quantile groups according to the BMI order (BMI: G1 < G2 < G3 < G4) (Fig. 8h and Supplementary Table 1). A total of 200 lipids, including 8 lysoplasmalogens and 59 plasmalogens, were identified with high confidence (Supplementary Data 3). ANOVA with Tukey's HSD post-hoc analysis revealed that 23 lipids showed significant differences (FDR < 0.05) among the four groups, including 1 lysoplasmalogen and 10 plasmalogens (Fig. 8i and Supplementary Fig. 22). Most significantly changed plasmalogens contained polyunsaturated fatty acids and showed a tendency towards higher levels in the higher body mass index (BMI) group (Supplementary Fig. 22 and Supplementary Data 3), as reported by Pietiläinen et al.[5]. Despite these changes, the highest BMI quartile had the lowest level of LPE P-18:0, which overall were negatively correlated with BMI (Fig. 8j), suggesting the clinical relevance of the beneficial effects of LPE P-18:0 treatment against obesity-related metabolic diseases.

## Discussion

Dysregulation of plasmalogen metabolism in adipose tissue has been implicated in the development of obesity-related metabolic diseases, yet little is known about the mechanisms regulating plasmalogen homeostasis in adipocytes or their impact on systemic metabolism. We observed that HFD causes an increase in RNA transcript and protein expression of TMEM86A, a putative lysoplasmalogenase in adipose tissue. We hypothesized that lysoplasmalogenase might be involved in plasmalogen metabolism in adipose tissue, and this could relate to obesity and metabolic imbalance and dysfunction. In this research, we characterized the effects of TMEM86A deletion in a genetic mouse model and the effects of overexpression of TMEM86A in adipocytes. Analysis of phospholipid composition by untargeted lipidomics demonstrated that TMEM86A overexpression in adipocytes significantly decreased lysoplasmalogens and plasmalogens. Conversely, TMEM86A KO in mouse adipocytes elevated the levels of lysoplasmalogens in adipose tissue. Furthermore, experiments with TMEM86A mutants suggested that the highly conserved aspartate residues (D82, D190) in YhhN proteins[6] are important for its lysoplasmalogenase activity. Together, these data indicate that TMEM86A acts as a bona fide lysoplasmalogenase.

Transcript levels of TMEM86B, a known lysoplasmalogenase[6], were considerably lower than those of TMEM86A in adipose tissue, suggesting that TMEM86A is a major lysoplasmalogenase in adipose tissue. TMEM86A expression was higher in WAT than in BAT. The finding that transcript levels of TMEM86A, but not TMEM86B were further increased in WAT by HFD feeding strongly suggested involvement of TMEM86A in adipose tissue remodeling in response to nutritional stimuli.

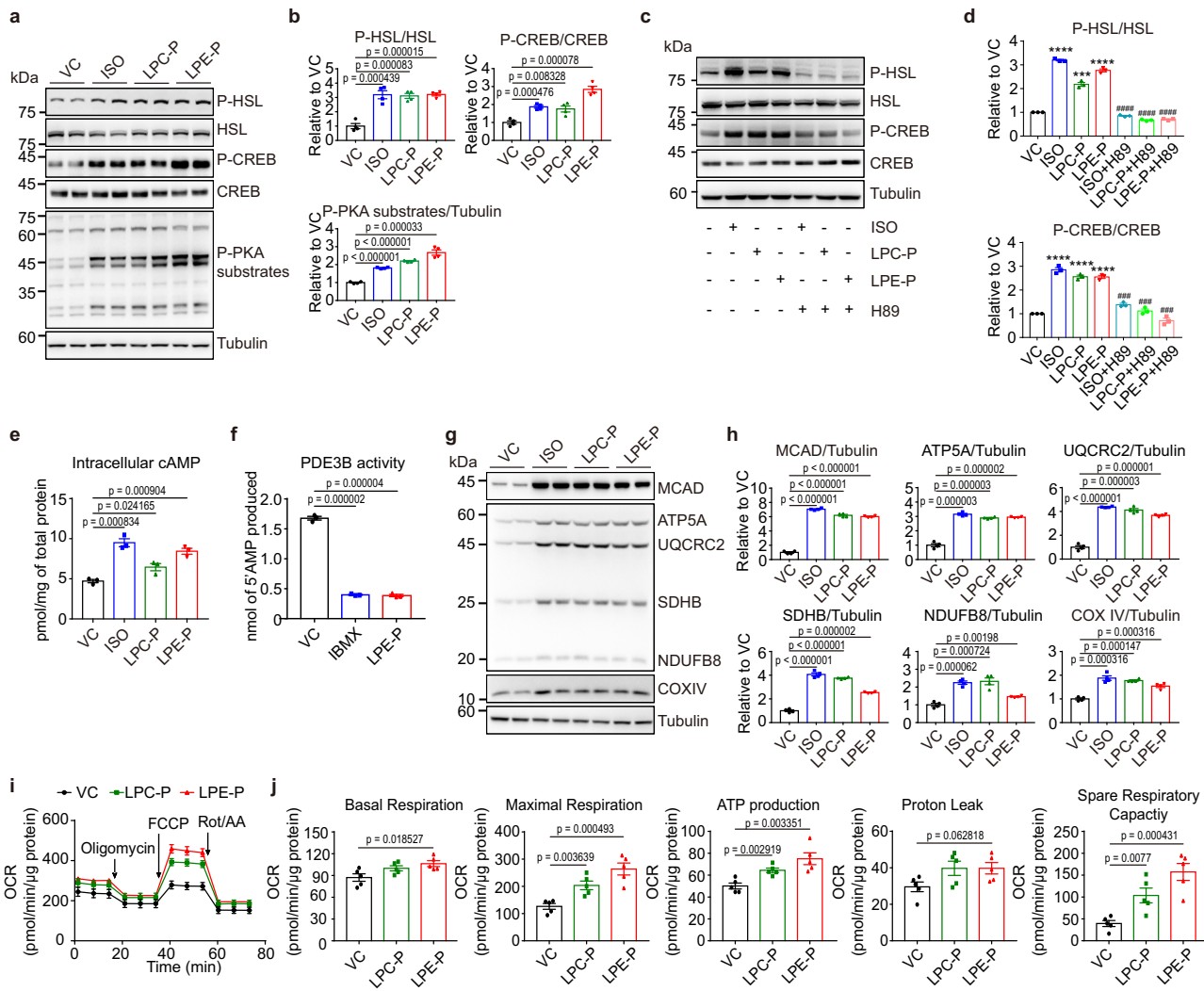

**Fig. 7 In vitro LPE P-18:O treatment elevates intracellular cAMP levels and activates PKA signalling via inhibition of PDE activity. a, b** Immunoblot analysis of C3H10T1/2 adipocytes treated with vehicle control (VC), LPC P-18:0 (LPC-P, 10 μM) or LPE P-18:0 (LPE-P, 10 μM) for 24 h or isoproterenol (ISO, 10 μM) for 4 h. n = 4. **c, d** Immunoblot analysis of C3H10T1/2 adipocytes treated with VC, ISO (10 μM), LPC-P (10 μM), LPE-P (10 μM), ISO (10 μM) + H89 (PKA inhibitor, 50 μM), LPC-P (10 μM) + H89 (50 μM), or LPE-P (10 μM)+H89 (50 μM). n = 3. (p-HSL/HSL VC vs. ISO: p < 0.000001; VC vs. LPC-P: p = 0.000128; VC vs. LPE-P: p = 0.000009; ISO vs. ISO + H89: p < 0.000001; LPC-P vs. LPC-P + H89: p = 0.000056; LPE-P vs. LPE-P + H89: p = 0.000006)(p-CREB/CREB VC vs. ISO: p = 0.000035; VC vs. LPC-P: p = 0.000018; VC vs. LPE-P: p = 0.000019; ISO vs. ISO + H89: p = 0.000172; LPC-P vs. LPC-P + H89: p = 0.000143; LPE-P vs. LPE-P + H89: p = 0.000109). **e** Intracellular cAMP levels in C3H10T1/2 adipocytes treated with VC, LPC-P (10 μM), or LPE-P (10 μM) for 24 h or ISO (10 μM) for 4 h. n = 3. **f** Levels of 5' AMP released by PDE3B incubated with VC, IBMX (40 μM), or LPE-P (10 μM) for 90 min. n = 3. **g, h** Immunoblot analysis of C3H10T1/2 adipocytes treated with VC, LPC-P (10 μM) or LPE-P (10 μM) for 48 h or isoproterenol (ISO, 10 μM) for 24 h. n = 4. **i, j** Oxygen consumption rate measurement in C3H10T1/2 adipocytes treated with VC, LPC-P (10 μM) or LPE-P (10 μM) for 24 h. n = 5. Each point represents a biological replicate. Data are presented as mean ± SEM. Statistical significance was determined by the unpaired, two-tailed t-test in **b, d, e, f, h** and **j**. * represents statistical analysis between C3H10T1/2 adipocytes treated with VC and C3H10T1/2 adipocytes treated with either ISO, LPC-P, or LPE-P. # represents statistical analysis between C3H10T1/2 adipocytes treated with either ISO, LPC-P, or LPE-P and C3H10T1/2 adipocytes treated with either ISO + H89, LPC-P + H89, or LPE-P + H89. Source data are provided as a Source Data file.

The lipidomics analyses in TMEM86A AKO mice and in the overexpressing adipocytes showed that TMEM86A lysoplasmalogenase controls the levels of both lysoplasmalogens and plasmalogens. Consistently, previous work reported that overexpression of TMEM86B decreases both lysoplasmalogens and plasmalogens in HEK cells[6]. While the mechanistic connection between plasmalogen and lysoplasmalogen levels is not known, it is possible that loss of lysoplasmalogens reduces the abundance of acyl groups available for transacylation of lysoplasmalogen to plasmalogens. It also seems likely that reduced levels of lysoplasmalogens and plasmalogens indirectly affect the levels of other lipid species within the plasmalogen metabolic

pathway. Further analysis of enzyme activity and structural demonstration of interactions between lipid substrates and TMEM86A protein would be informative for a comprehensive understanding of its physiological function.

In this study, we demonstrated that elevation of lysoplasmalogen levels by TMEM86A deficiency facilitated PKA signalling. In turn, enhanced cAMP-dependent phosphorylation of key metabolic enzymes and transcription factors, such as HSL and CREB, promoted expression of mitochondrial electron transport enzymes and expanded oxidative catabolism of adipocytes. Mechanistically, we found that LPE P-18:0 inhibited PDE3B, the major enzyme that degrades cAMP in adipocytes. Previous

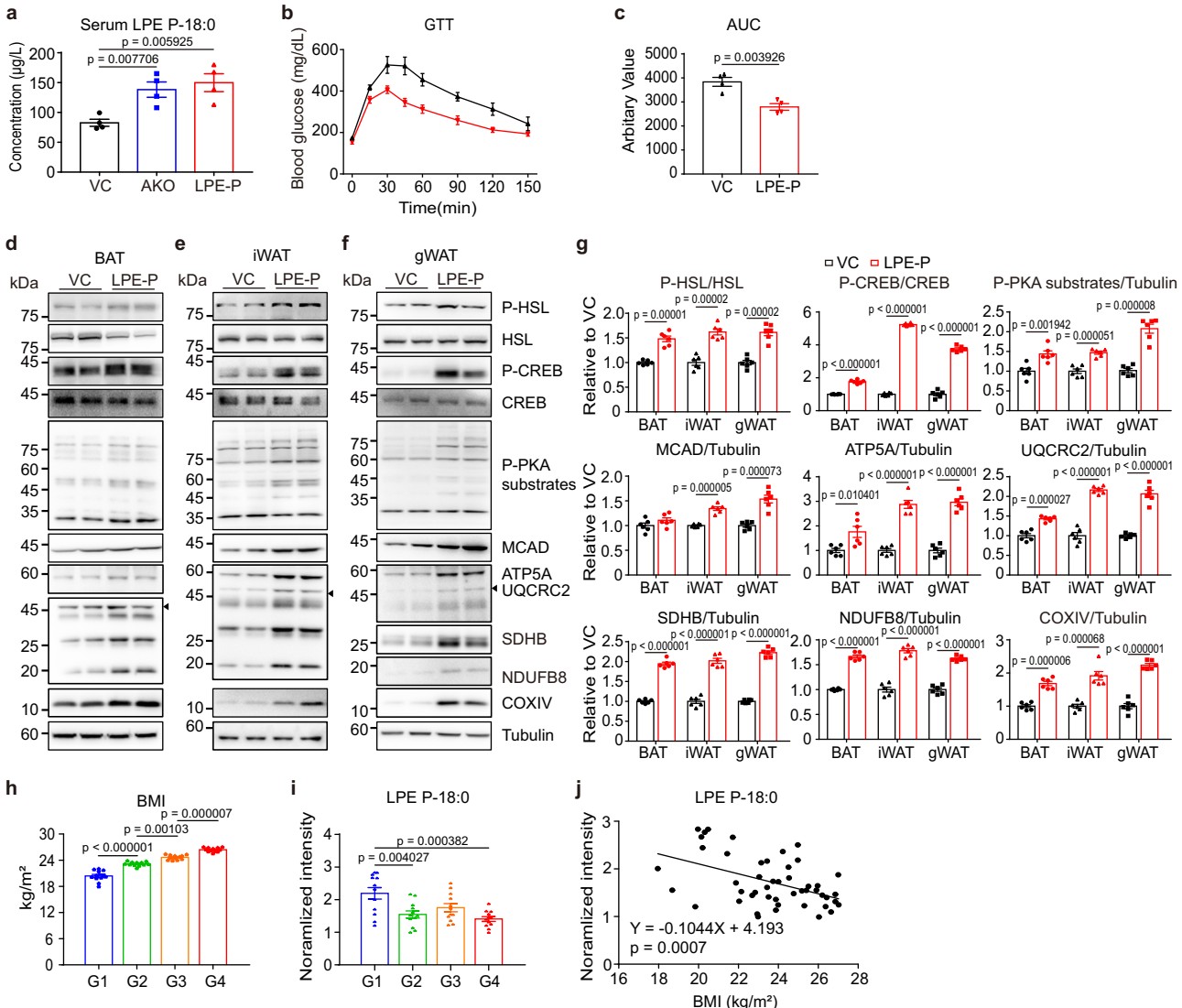

**Fig. 8 LPE P-18:0 treatment facilitates mitochondrial oxidative metabolism and protects against diet-induced obesity. a** Concentration of LPE P-18:0 (LPE-P) in serum of TMEM86A AKO mice or WT mice treated with either VC or LPE-P for 2 weeks (i.p. 200 µg/kg/day). n = 4. **b, c** Intraperitoneal glucose tolerance test of WT mice fed a HFD treated intraperitoneally with VC or LPE-P for 2 weeks. n = 4. **d–g** Immunoblot analysis of BAT, iWAT and gWAT of WT mice fed a HFD treated with VC or LPE-P for 2 weeks (i.p. 200 µg/kg/day). n = 6. **h** Grouping of 48 individuals in quartile according to body mass index (BMI). **i** Normalized intensities of lysoplasmalogen LPE P-18:0 in human subcutaneous adipose tissue (Intensity was normalized by median, *FDR < 0.05). n = 48. **j** Correlation analysis of normalized intensity of LPE P-18:0 with BMI. n = 48; linear regression of correlation; R = −0.4717, p = 0.0007. Each point represents a biological replicate. Data are presented as mean ± SEM. Statistical significance was determined using the unpaired, two-tailed t-test in **a**, **c**, and **g** and using ANOVA (one-way, Tukey's HSD post-hoc analysis) in **h** and **i**. Source data are provided as a Source Data file.

studies indicated that LPC P- increases PKA activity by direct action on the catalytic subunit[21]. This study also reported LPE P- or other lysophospholipid classes do not activate PKA, indicating the specificity of PKA for lysophospholipids having choline polar head groups. In contrast, we found that LPE P-18:0 increased PKA activity in adipocytes by inhibiting PDE3B, thereby increasing cAMP levels. Further investigation is required to understand the molecular mechanisms of LPE P-18:0-induced inhibition of PDE3B activity.

Our current study demonstrated that lysoplasmalogenase deficiency increased plasmalogen content, as well as lyso-plasmalogen. Plasmalogen metabolism generates diverse lipid mediators; thus, several studies have investigated the correlation between plasmalogen levels and metabolic disease states. For example, a previous study demonstrated that PC O-16:0/18:1 and PC O-16:0/18:2 is inversely associated with waist circumference[28].

In particular, PC O-16:0/18:1 had a positive relationship with c-peptide and adiponectin, which suggested its association with improved insulin signalling and glucose metabolism[28]. An in vivo study performed by Park et al. reported that adipocyte-specific deletion of *Pex16*, a crucial factor for peroxisome biogenesis that is responsible for plasmalogen synthesis, led to the prevention of mitochondrial fission, impaired thermogenesis, and reduced peroxisomes in adipocytes[7]. Supplementing PEX16 KO mice with plasmalogens rescued mitochondrial function and thermogenesis[7]. In contrast, independent studies have demonstrated that high plasmalogen and arachidonic acid content in the adipose tissue of individuals with obesity increases vulnerability to inflammation[5]. These seemingly contradictory results may be due to the complex nature of plasmalogen structure and metabolism, which are uniquely attributed to tissue types and developmental stages.

In the current study, we found that deletion of TMEM86A in adipocytes protects mice from obesity and insulin resistance in vivo. This was accompanied by improved glucose tolerance and insulin sensitivity and reduced pro-inflammatory macrophage polarization. Several mouse model studies have reported that increase in lipid catabolism and mitochondrial activity in the adipose tissue is beneficial in protection against HFD-induced insulin resistance and inflammation[29]. Thus, it is reasonable to consider that lysoplasmalogen-induced PKA signalling in the adipose tissue could be therapeutically targeted for overcoming obesity-induced metabolic dysfunction[29,30].

In summary, our data suggest the important role of TMEM86A-mediated lysoplasmalogen metabolism in adipose tissue function and indicate a potential therapeutic strategy for obesity-related metabolic diseases.

## Methods

**Ethics**. The research complies with all relevant ethical regulations. All the protocols of animal studies were reviewed and approved by the Institutional Animal Care and Use Committees of Seoul National University (SNU-190821-2, SNU-201123-2). All animal experiments were conducted according to the guidelines for humane care and use of laboratory animals (Ministry of Food and Drug Safety). All patients provided written informed consent, and the study was approved by the Institutional Review Board of Pusan National University Hospital (permit number: PNUH IRB 2101-019-099).

**Animals**. In vivo experiments were performed in Animal Center for Pharmaceutical Research at Seoul National University, where mice were housed at $22 \pm 1$ °C, $50\% \pm 5$ humidity, 12-h light/12-h dark cycle condition with free access to food and water. $Tmem86a^{flox/flox}$ mice were generated by crossing C57BL/6N-$Tmem86a^{tm1a(KOMP)Mbp}$ mice (Korea Mouse Phenotyping Center, #MOP1812011) with ACTB-FLPe (B6.Cg-Tg(ACTFLPe)9205Dym/J; JAX stock #005703). Adipoq-Cre (B6;FVB-Tg(Adipoq-cre)1Evdr/J; stock#028020) mice were then bred with $Tmem86a^{flox/flox}$ mice to generate the following genotypes: $Tmem86a^{flox/flox}$ without adipoq-Cre (WT), $Tmem86a^{flox/flox}$; adipoq-Cre (TMEM86A AKO)[31]. C57BL/6 mice (six weeks old, male) were purchased from Joongah Bio (Korea). Male WT and TMEM86A AKO mice were sacrificed at 8-16 weeks of age for in vivo experiments. For the diet-induced obesity model, mice at 8 weeks of age were fed with a high-fat diet (HFD) (60% fat, Research Diets, cat#D12492) for eight weeks. As a control chow diet, a standardized rodent pellet diet (NCD, Purina, cat#38057) was used. Energy expenditure was estimated by indirect calorimetry (PhenoMaster, TSE Systems, Germany). Body composition was measured by nuclear magnetic resonance (NMR) scanning EchoMRI-700 (Echo Medical Systems). To examine the effects of LPE P-18:0 treatment, mice were fed a HFD for 8 weeks and were intraperitoneally injected with LPE P-18:0 (Avanti Polar Lipids, 200 μg/kg/day, cat#852471 P) or vehicle control for 2 weeks.

**LPC P-18:0/LPE P-18:0 preparation**. For lipid solution, 95% methanol was added to the lipid at a concentration of 0.5 mg/mL. The mixture was sonicated until the lipid is suspended, and transferred to a glass container with a teflon closure. Methanol was removed by dry nitrogen and the glass vessels were rotated during evaporation to make a thin film on the tube wall. The lipid film was stored at −80 °C. To make lipid solution, the container was placed at room temperature before opening. Fatty acid free bovine serum albumin (Sigma, cat#A8806-1G, 4 mg/mL) was then added to make a 100 μM lipid solution. The solution was vortexed and sonicated occasionally until the solution becomes clear.

**Cell culture**. HEK293T cells were obtained from American Type Culture Collection (ATCC, CRL-3216) and cultured in Dulbecco's Modified Eagle Medium (DMEM, Welgene, cat#LM001-07) supplemented with 10% fetal bovine serum (FBS, Gibco, cat#16000044) and 1% penicillin/streptomycin (P/S, Welgene, cat#LS202-02). C3H10T1/2 (ATCC, CCL-226) cells were used to conduct in vitro adipocytes experiments and cultured in DMEM supplemented with 10% FBS and 1% P/S at 37 °C in a humidified atmosphere with 5% CO$_2$. To generate fully differentiated C3H10T1/2 adipocytes, confluent C3H10T1/2 cells were exposed to bone morphogenetic protein 4 (R&D systems, 20 ng/mL, cat#314-BP-050) for 2 days and induced to differentiation medium containing IBMX (Sigma, 0.5 mM, cat#I5879-1G), dexamethasone (Cayman, 1 μM, cat#11015), insulin (Sigma, 10 μg/mL, cat#I9278), indomethacin (Cayman, 0.125 mM, cat#70270) and triiodothyronine (T3, Cayman, 1 nM, cat#16028) for 3 days followed by exposure to maintenance medium containing insulin (1 μg/mL) and T3 (1 nM) for 3 days.

To assess the in vitro effects of lysoplasmalogen, fully differentiated C3H10T1/2 adipocytes were treated with either vehicle controls, LPE P-18:0 (10 μM) or LPC P-18:0 (10 μM, cat#852465 P), or isoproterenol (10 μM, Sigma, cat#I5627-5G) for indicated times.

**Primary cell culture**. BAT, iWAT, and gWAT from 8 weeks old male WT and TMEM86A AKO mice were used for primary cell studies. Adipose tissues were isolated, washed, minced, and digested with collagenase type I (Gibco, 2 mg/mL, cat#17100-017) in KRBB buffer containing 3% BSA at 37 °C. Fully dissociated preparations were then passed through a 100 μm cell strainer (Corning) and centrifuged at $300 \times g$ for 5 min. Floating adipocytes were collected for further analysis. Red blood cells were lysed with RBC lysis buffer (15.5 mM NH$_4$Cl, 1.2 mM NaHCO$_3$ and 50 mM EDTA, pH 7.4) and pellets were then washed with DMEM containing 20% FBS and 1% P/S for two times. Stromal vascular fraction (SVF) was filtered through 40 μm strainer (Corning) and distributed onto the plate-format of interest. The isolated adipocyte precursors were washed with PBS and the media was replaced each day. When the adipocyte precursors were 100% confluent, cells were induced to differentiation medium containing IBMX (0.5 mM), dexamethasone (1 μM), insulin (10 μg/mL), and indomethacin (0.2 mM) for 3 days and then induced to maintenance medium containing insulin (10 μg/mL) for 3 days. Fully differentiated primary adipocytes were further analyzed with immunoblot and oxygen consumption measurements.

**Lentiviral vector-mediated TMEM86A overexpression**. For overexpression of TMEM86A in C3H10T1/2 adipocytes, mouse $Tmem86a$ (NM_026436) ORF Clone was purchased from Origene. Lentiviral transfer plasmids $Tmem86a$ ORF were generated by inserting C-terminus Myc-tagged $Tmem86a$ ORF clone to the lentiviral transfer vector pLenti-EF1a-C-mGFP-P2A-Puro lentiviral vector (Origene, cat#PS100121) using AsiSI and MluI. To produce lentivirus, HEK293T cells were transfected with transfer plasmids (5 μg), psPAX2 (Addgene, 3.75 μg, cat#12260), and pMD2.G (Addgene,1.25 μg, cat#12259) in 4:3:1 ratio using jetPRIME transfection reagent (Polyplus, cat#114-15). The medium containing viral particles was collected after 48 h of transfection, centrifuged, and filtered through 0.45-nm pore (Sartorius). C3H10T1/2 cells were transduced with the viral supernatants containing polybrene (Santa Cruz, 8 μg/mL, cat#sc-134220) for 48 h and were replaced with growth medium containing puromycin (Sigma, 2 μg/mL, cat#P8833) for selection.

**Mini Gs expression**. DNA fragment encoding nuclear export sequence (MLQNELALKLAGLDINKT)-linker A (GGSG)-mVenus-linker B (GGGGS)-mini Gs393 sequences[27] were synthesized from Bio Basic Inc. The DNA fragments were amplified with PCR by adding EcoRI site to 5'-terminus and XbaI site to 3'-terminus. Then, PCR-amplified DNA constructs were digested with EcoRI/XbaI and subcloned into pLVX-EF1a-mCherry-N1 (Takara, cat#631986) by substituting mCherry sequence (hereinafter referred to as 'mini G plasmid'). To study the interaction between G protein and GPCR, mini G plasmid and pcDNA3 Flag beta-1-adrenergic-receptor (Addgene, cat#14698) were co-transfected in HEK293T with jetPRIME transfection reagent in a 1:1 ratio. Cells were observed with confocal microscope 24 h after transfection.

**Site-directed mutagenesis**. To generate mutant $Tmem86a$ sequence, EZchange Site-directed Mutagenesis kit (Enzynomics, cat#EZ004S) was used following the manufacturer's manual. nPfu-Forte DNA polymerase, mouse $Tmem86a$ (NM_026436) ORF Clone, and the indicated primers (D82A forward: 5'-CTG CTG TGG GTG CTG CCT TCC TCA TC-3', D82A reverse: 5'-AAA AGA CAA GTC CCA CGA AGA TGA GGG-3'; D190A forward: 5'-TCA TCC TCT CGG CCC TGA CCA TCG C-3', D190A reverse: 5'-AGA GCA GAG CAC CAC CGC CGG-3') were used to generate the potential active site mutant sequences. As a result, Asp 82 was substituted by Ala 82 in D82A mutation and Asp 190 was substituted by Ala 190 in D190A mutation. Mutations were confirmed by DNA sequencing. To assess the effect of $Tmem86a$ mutation in lysoplasmalogenase activity, each mutant $Tmem86a$ plasmid (1 μg) was transiently transfected in HEK293T cells using jetPRIME transfection reagent. HEK293T cells were treated with LPE P-18:0 24 h after transfection and the media was collected at 0, 2, 4, 6, 8, and 12 h after LPE P-18:0 treatment to measure LPE P-18:0 consumption levels. For immunoblot analysis, protein was extracted 48 h after transfection.

**Magnetic cell sorting**. F4/80+, PDGFRα+, and F4/80-PDGFRα- cells were further isolated from SVF by magnetic cell sorting (MACS). Briefly, SVF was incubated with anti-F4/80-FITC (Biolegend, 1:50) for 1 h at 4 °C followed by anti-FITC-microbeads (Miltenyi Biotec, 1:10, cat#130-048-701) incubation for 30 min at 4 °C. After magnetic separation of F4/80+ cells, PDGFRα+ cells were isolated by using PDGFRα MicroBead kit (Miltenyi Biotec, cat#130-101-502) for magnetic separation.

**Measurement of oxygen consumption rates**. Oxygen consumption rates (OCRs) were measured by the Seahorse XF Analyzers (Agilent)[32,33]. Primary cultured adipocytes from gWAT of WT or TMEM86A AKO, differentiated C3H10T1/2 adipocytes overexpressing Mock or TMEM86A, and differentiated C3H10T1/2 adipocytes treated with either vehicle, LPC P-18:0 (10 μM), or LPE P-18:0 (10 μM) were washed and maintained in assay medium (XF DMEM Base Medium, pH 7.4 (Agilent, cat#103575-100)) supplemented with D-(+)-glucose (25 mM) and L-glutamine (4 mM) at 37 °C. XFp Cell Mito Stress Test Kit (Agilent, cat#103010-100) was used with the following concentrations: 2.5 μM of oligomycin, 0.5 μM of

carbonyl cyanide-4-(trifluoromethoxy) phenylhydrazone (FCCP), and 0.5 μM Rotenone/Antimycin A. Basal and maximal OCRs were calculated by subtraction of non-mitochondrial respiration. ATP production-related OCRs were calculated by subtracting the oligomycin A-induced OCR from the basal OCR. Spare respiratory capacity was calculated by subtracting Basal OCR from Maximal OCR. Proton leak was calculated by subtracting non-mitochondrial respiration from the oligomycin A-induced OCR. OCRs were normalized with protein concentrations. Agilent Wave software (version 2.6.0.31) was used to analyze the data.

**Quantification of intracellular cAMP levels**. To measure intracellular cAMP levels from differentiated C3H10T1/2 adipocytes treated with either vehicle, LPC P-18:0 (10 μM), LPE P-18:0 (10 μM), or isoproterenol (10 μM), direct cAMP ELISA kit (Enzo, cat#ADI-900-066) was used following the manufacturer's instruction. Briefly, C3H10T1/2 adipocytes were collected after incubation with 0.1 M HCl for 10 min and centrifuged. The collected samples and standards, which were acetylated and added to wells coated with a goat-rabbit IgG antibody, were then incubated with cAMP conjugated to alkaline phosphatase and rabbit polyclonal antibody to cAMP for 2 h at room temperature on a plate shaker. After three times of washing, a solution of p-nitrophenyl phosphate was added and incubated at room temperature for 1 h followed by addition of a solution of trisodium phosphate in water. The plate was measured at 405 nm and the calculated absorbance was converted to pmol/ml of cAMP using AssayFitPro (v1.31) and normalized with protein concentration measured by Pierce BCA Protein Assay Kit (Thermo Fisher Scientific, cat#23225).

**PDE3B activity assay**. To examine the inhibitory effect of LPE P-18:0 on cyclic nucleotide PDE, Cyclic Nucleotide Phosphodiesterase Assay Kit (Enzo, cat#BML-AK800-0001) was applied following the protocol provided by the manufacturer. Briefly, the mixtures of cAMP substrate (200 nM), assay buffer, 5'-nucleotidase (1 kU/μL), and PDE3B (40 pg/reaction, BPS Bioscience, cat#60031) with LPE P-18:0 (10 μM) or IBMX (40 μM) were incubated in a microtiter plate for 0, 45, or 90 min at 30 °C. Incubation time was determined by a linearity test indicating the initial rate of the enzyme. Next, BIOMOL Green reagent was added and incubated for 30 min at 30 °C to terminate reactions. The plate was measured at 620 nm and the measure absorbance was converted to the nmol of 5'-AMP produced by using the equation for best-fit line obtained from the standard curve.

**Ex vivo triphenyl tetrazolium chloride assay**. Ex vivo electron transport activity related to mitochondrial oxidative phosphorylation was evaluated by monitoring the reduction of 2,3,5-Triphenyltetrazolium chloride (TTC, Sigma, cat#T8877)[34]. Approximately 10 mg of each adipose tissue depot was incubated in PBS containing 2% TTC for 15 min at 37 °C followed by fixation with 10% formalin for 30 min at room temperature. Each sample was then transferred to 95% ethanol to extract the formazan product and incubated overnight at 4 °C. The absorbance of the extracted solution was measured at 485 nm using MultiSkan GO spectrophotometry and the values were normalized by tissue weights.

**Measurement of liver triglyceride content**. To measure hepatic TG content, Triglyceride Colorimetric Assay Kit (Cayman, cat#10010303) was used following manufacturer's instruction. Briefly, 100 mg of liver was minced and homogenized in diluted NP40 Substitute Assay Reagent containing protease inhibitor. Tissue homogenates were then centrifuged at $10,000 \times g$ for 10 min at 4 °C. Samples were diluted and then incubated with Enzyme Mixture solution for 15 min at room temperature to initiate the reaction and the absorbance was measured at 540 nm using MultiSkan GO spectrophotometry.

**Glucose tolerance test (GTT)**. Mice were fasted for 16 h prior to the measurement of blood glucose levels. D-(+)-glucose (Sigma, 2 g/kg body weight, cat#49139) was injected intraperitoneally and blood glucose levels were measured from tail vein blood samples at indicated time using Gluco Doctor Top meter (Allmedicus, cat#AGM-4100) and appropriate glucose indicator strips.

**Insulin tolerance test (ITT) and acute insulin treatment**. For insulin tolerance test, mice were fasted for 6 h prior to the test and given insulin (Sigma, 0.75 units/kg body weight, cat#91077C-1G) by intraperitoneal injection, and then blood glucose level was measured at indicated time intervals.

To examine insulin signalling, mice were anesthetized and insulin (0.75 units/kg body weight) was injected into the inferior vena cava and adipose tissues were harvested 10 min after the injection.

**Flow cytometric analysis**. SVF was obtained as described above and stained with fluorescence labeled primary antibodies or control IgG for 20 min at room temperature. CD45 (Biolegend, 30-F11, cat#103116) antibody was used to label leukocyte population from SVF. Of the leukocytes, CD64 (Biolegend, FcγRI, X54-5/7.1, cat#139311) and CD11b (Biolegend, M1/70, cat#101206) antibodies were used to identify total macrophage population and CD11c (Biolegend, N418, cat#117310) antibody and CD206 (Biolegend, MMR; C068C2, cat#141717) antibody were used to classify M1-like and M2-like macrophages, respectively. Unstained and single

stained controls were used for compensation. LSRFortessa X-20 Flow Cytometer (BD Biosciences) was used to analyze the samples and the data was acquired by BD FACSDiva 8.0 software which then was further analyzed by Flowjo software (version 10.5.3; TreeStar). The gating strategy used in the analysis is included in Supplementary Fig. 15b.

**RNA sequencing**. TRIzol reagent (Thermo Fisher Scientific, cat#15596018) was used for tissue total RNA extraction according to the manufacturer's protocol. RNA integrity number (RIN), rRNA ratio, and concentration of samples were verified on an Agilent Technologies 2100 Bioanalyzer (Agilent Technology) using a DNA 1000 chip. For RNA-seq analysis, cDNA libraries were constructed with the TruSeq mRNA Library Kit using 1 μg of total RNA. The total RNA was sequenced by the NovaSeq 6000 System (Macrogen). The RNA-seq data were deposited in Gene Expression Omnibus (GEO) (accession number GSE182930).

**Quantitative PCR and western blot analysis**. Total RNA from adipose tissue was extracted using TRIzol reagent following the manufacturer's manual. cDNA was synthesized using High-Capacity cDNA Reverse Transcription kit (Applied Biosystems, cat#4368814) and qPCR reaction was run with iQ SYBR Green Supermix (Bio-Rad, cat#170-8884) in Bio-Rad CFX Connect Real Time PCR Detection system. The acquired data were further analyzed with Maestro 1.1 software (version 4.1.2433.1219). Relative expression levels of each gene were calculated using the 2-ΔCt method. Peptidylprolyl isomerase A (*Ppia*) was used as a housekeeping gene and primers used for qPCR are listed in Supplementary Table 2.

BAT, iWAT, and gWAT depots of WT and TMEM86A AKO mice were homogenized in PRO-PREP Protein Extraction Solution (iNtRON Biotechnology, cat#17081) containing SIGMAFAST Protease Inhibitor Cocktail (Sigma, cat#S8820) and PhoSTOP phosphatase inhibitors (Roche, cat#4906845001) using tacoPrep Bead Beater homogenization system. Differentiated adipocytes were lysed in RIPA Lysis and Extraction Buffer (Thermo Fisher Scientific, cat#89900) containing SIGMAFAST Protease Inhibitor Cocktail (Sigma) and PhoSTOP phosphatase inhibitors (Roche). Extracted protein concentration was quantified by BCA assay and measured by spectrophotometry (MultiSkan GO, Thermo Fisher Scientific) at 562 nm using SkanIt software ver 5.0. Proteins were denatured in SDS sample buffer at 95 °C for 5 min and separated on SDS-PAGE gel and then transferred to PVDF membrane (Bio-Rad). For detection of TMEM86A, proteins were denatured in SDS sample buffer at 40 °C for 30 min. The membrane was blocked with 5% non-fat dry milk or BSA in TBST (Tris-buffered saline with 0.1% Tween 20) and incubated with primary antibody at 4 °C overnight followed by horseradish peroxidase-conjugated secondary antibody incubation at room temperature for 1 h. Primary antibodies against TMEM86A (customized, 1:1000), COXIV (cat#4850 S, 1:1000), phospho-HSL (Ser660, cat#45804 S, 1:1000), HSL (cat#4107 S, 1:1000), phospho-CREB (cat#9198 S, 1:1000), CREB (cat#9197 S, 1:1000), phospho-PKA substrates (cat#9621 S, 1:1000), F4/80 (cat#30325 S, 1:1000), Tubulin (cat#2148 S, 1:1000) were purchased from Cell Signaling Technology, B-actin (cat#sc-47778, 1:1000), MCAD (cat#sc-365030, 1:1000), phospho-AKT1/2/3 (cat#sc-514032, 1:1000), and AKT1/2/3 (cat#81434, 1:1000) were purchased from Santa Cruz Biotechnology, PDE3B (cat#14-1973-82, 1:500) and Horseradish Peroxidase-conjugated Goat anti-Rabbit IgG (H + L) Secondary Antibody (cat#31460, 1:5000) were purchased from Invitrogen, Horseradish Peroxidase-conjugated Goat anti-mouse IgG, light chain specific Secondary Antibody (cat#115-035-174, 1:1000) was purchased from Jackson ImmunoResearch and Total OXPHOS rodent cocktail (cat#110413, 1:1000) was purchased from Abcam. Protein expression was detected with Fusion Solo chemiluminescence imaging system (Vilber Lourmat) and analyzed with EvolutionCapt software (version 17.03). Immunoblots were quantified with National Institutes of Health ImageJ software (version 1.52a). Full immunoblot membrane images are provided in the Source Data and Supplementary Information.

**Live-cell imaging and immunofluorescence microscopy**. For ER and lipid staining, live C3H10T1/2 cells and adipocytes were incubated in growth media containing ER-Tracker red (Invitrogen, 1 μM, cat#E34250) and HCS LipidTOX™ Deep Red Neutral Lipid Stain (Invitrogen, 1:2500, cat#H34477) for 1 h at 37 °C and 5% $CO_2$. For live-cell imaging, the growth media containing the probes was replaced with probe-free fresh media and cells were directly observed using Zeiss confocal microscope (LSM800) with live-cell chamber and analyzed with Zen software (ZEN 3.0 Blue edition). Cells were fixed in 4% paraformaldehyde for 2 min at 37 °C without permeabilization after ER tracker staining. For immunofluorescence staining, cells were fixed in 4% paraformaldehyde for 20 min, permeabilized in 0.2% Triton-X100 for 5 min, and then incubated in blocking buffer (PBS containing 3% BSA) for 1 h at room temperature. Anti-PDI antibody (Cell Signaling, 1:200, cat#3501) was applied overnight at 4 °C. Thereafter, Alexa Fluor 594 (Invitrogen, 1:500, cat#A11012) was applied for 1 h at room temperature. Nuclei were counter-stained with DAPI.

**Histology and immunohistochemistry**. BAT, iWAT, and gWAT were isolated and fixed in 10% formalin (Sigma, cat#HT50128) for 24 h at 4 °C and then embedded in paraffin. 5 μm paraffin sections were prepared from the paraffin embedded tissue blocks. The sections were deparaffinized with the following steps:

$2 \times 10$ min in xylene, 10 min in 50:50 xylene:ethanol, 5 min in 100% ethanol, 5 min in 95% ethanol, 5 min in 70% ethanol. The slides were rinsed with distilled water, boiled for 10 min in citrate buffer (pH 6.0) for antigen-retrieval, and cooled to room temperature. For H&E staining, deparaffinized sections were stained with ClearView Staining Hematoxylin (BBC Biochemical, cat#MA010081) and Eosin Y Alcoholic solution (BBC Biochemical, cat#3610). The images were obtained with Nikon Elements (NIS BR Analysis ver 5.10.00). For immunofluorescence staining, blocking was performed with PBS containing 3% BSA for 30 min at room temperature. Anti-F4/80 antibody (Cell Signaling) was diluted in 3% BSA in PBS (1:400) and applied onto the slides for incubation overnight in a humidified chamber. Subsequently, the slides were washed three times and incubated with goat anti-rabbit IgG cross-adsorbed secondary antibody, Alexa Fluor 488 (Invitrogen, 1:500, cat#A11008) for 1 h at room temperature. Finally, the slides were mounted using PermaFluor Mountant (Thermo Fisher Scientific, cat#TA-030-FM) with DAPI. Images were acquired with Zeiss confocal microscope (LSM800) and analyzed with Zen software (version 3.0).

**Transmission electron microscopy (TEM)**. For TEM imaging, small pieces of minced adipose tissue (1–2 mm$^3$) were incubated in 2% paraformaldehyde and 2.5% glutaraldehyde in 0.1 M sodium cacodylate buffer (pH 7.2). Tissues were post-fixed in 1% osmium tetroxide (OsO$_4$) for 2 h, dehydrated using ethanol gradient, and embedded into Spurr's Resin (Electron Microscopy Sciences). Samples were then sectioned at 60 nm with Ultramicrotome (Leica) and subsequently transferred to nickel grids. Images were collected under Talos L120C cryo-TEM (FEI) transmission electron microscope at the Nanobio Imaging Center, Seoul National University.

**Lipid extraction**. For mouse and human adipose tissue, approximately 20 mg of tissue samples were homogenized by bead beater (Bertin Technologies) at 7200 rpm for 30 s, 2 cycles. To remove the excess amount of neutral lipids in adipose tissue, the three-phase liquid extraction[25] with slight modification was applied to the homogenate. Briefly, 480 µL of water, 360 µL of acetonitrile, 480 µL of methyl acetate, and 480 µL of n-hexane were added to homogenate followed by vortexing and 1 h shaking at 4 °C. The samples were centrifuged for 10 min at $16,000 \times g$ and the middle layer was collected. Due to the excess of neutral lipid left, re-extraction was conducted by transferring the middle layer to a fresh tube and 480 µL of n-hexane was added and vortexed. Finally, the mixture was centrifuged for 10 min at $16,000 \times g$ and the bottom phase was transferred to a fresh tube and dried under nitrogen gas flow.

For adipocyte lipid analysis, a pellet from $1 \times 10^6$ cells was extracted using the Matyash method[35] with slight modification. Briefly, 300 µL of 4 °C methanol was added to the pellet followed by the addition of 250 µL of 4 °C water. After vortexing of the mixture, two cycles of freezing and thawing were conducted for cell lysis. Then, 1000 µL of 4 °C MTBE was added followed by vortexing and 1 h shaking at 4 °C. The sample was centrifuged for 10 min at $16,000 \times g$ and the upper organic layer was collected. Finally, it was dried under nitrogen gas flow.

LPC 12:0, LPE 17:1, PC 20:0, PE 20:0, PC P-36:1(d9), and PE P-36:1(d9) were diluted in extraction solvent and used as internal standards to check the reliability of the data. In addition, quality control (QC) sample was pooled by mixing the same portion from every sample. The pooled QC samples were injected multiple times for system conditioning at the beginning of the sequence[36], signal correction throughout the sequence, and acquisition of the MSMS spectra with Data-dependent acquisition[37]. Acid hydrolysis of lipid extract was conducted with some pooled QC samples for the discrimination of the plasmanyl phospholipids and plasmenyl phospholipids (plasmalogen)[38]. The dried QC samples in the vials were exposed to HCl fume 5 min followed by ventilation under nitrogen gas flow. All dried samples were kept in the −80 °C freezer before analysis. Dried lipid extracts were resuspended in 120 µL of methanol:toluene (9:1, v/v) and kept at 4 °C in an autosampler during analysis.

**Sample preparation for quantification of LPE P-18:0 in mouse serum**. For the quantification of LPE P-18:0 in mouse serum, 50 µL of mouse serum was extracted using the Matyash method. Briefly, 300 µL of 4 °C methanol was added to the serum followed by addition of 1000 µL of 4 °C MTBE. After vortexing, mixture was shaken at 4 °C for 1 h. Then, 250 µL of 4 °C water was added and vortexed. The mixture was centrifuged for 10 min at $16,000 \times g$ and the upper organic layer was collected. The upper organic layer was dried under nitrogen gas flow. In addition, LPE 17:1 was diluted in methanol before extraction and used as internal standard. Finally, dried lipid extracts were resuspended in 100 µL of methanol:toluene (9:1, v/v) and kept at 4 °C in an autosampler during analysis.

**Sample preparation for extracellular LPE P-18:0 analysis**. For the determination of LPE P-18:0 in culture medium of HEK293T and C3H10T1/2 cells, 60 µL of 100% methanol was added to 15 µL of medium, followed by 10 min incubation on ice. After vortexing, the mixture was centrifuged ($16,000 \times g$, 20 min, 4 °C), and the extracts of cell culture were analyzed by LC-MS.

**LC-MS conditions for lipidomics and LPE P-18:0 analysis**. For lipid analysis, LC-MS conditions were adopted from a previous publication[39]. 5 µL of the extracts

were injected into the Vanquish UHPLC system (Thermo Scientific Fisher) coupled with an Acquity UPLC CSH C18 column (100 × 2.1 mm, 1.7 µm) connected with an Acquity UPLC CSH C18 VanGuard pre-column (5 × 2.1 mm, 1.7 µm) (Waters, MA, USA). The mobile phase A was 0.1% formic acid and 10 mM ammonium formate in 60:40 (v/v) of acetonitrile:water, and the mobile phase B was 0.1% formic acid and 10 mM ammonium formate in 90:10 (v/v) of 2-propanol:acetonitrile. The column temperature was set to 65 °C, and the gradient elution was conducted as following at a flow rate of 0.6 mL/min: 0 min, 15% of phase B; 0–2 min, 30% of phase B; 2–2.5 min, 48% of phase B; 2.5–11 min, 82% of phase B; 11–11.5 min, 99% of phase B; 11.5–12 min, 99% of phase B; 12–12.1 min, 15% of phase B; 12.1–15 min, 15% of phase B; 4 min of post-run. Lipidomics data were acquired by Full MS mode for quantification; Data-dependent acquisition (DDA) mode with QC sample was used for lipid identification using Thermo Scientific Q Exactive Plus Hybrid Quadrupole-Orbitrap mass spectrometer and Xcalibur software (Thermo Scientific Fisher).

For the analysis of LPE P-18:0 in mouse serum, 5 µL of the extracts was injected into the 1290 UHPLC (Agilent) coupled with an Acquity UPLC CSH C18 column (100 × 2.1 mm, 1.7 µm) connected with an Acquity UPLC CSH C18 VanGuard pre-column (5 × 2.1 mm, 1.7 µm) (Waters, MA, USA). LC condition was the same as the above condition. The MS acquisition was operated at MRM mode using Agilent 6460 QqQ (Agilent) and MassHunter Workstation Data Acquisition software (Agilent). Authentic LPE P-18:0 standard was used for optimization of MRM transition and MS parameters.

For the analysis of LPE P-18:0 in cell culture medium, the chromatographic separation was performed as previously described[40]. 5 µL of medium extracts was injected into the Vanquish UHPLC system (Thermo Scientific Fisher) coupled with an Acquity UPLC BEH Amide column (150 × 2.1 mm, 1.7 µm) connected with an Acquity UPLC BEH Amide VanGuard pre-column (5 × 2.1 mm, 1.7 µm) (Waters, MA, USA). The mobile phase A was 0.125% formic acid and 10 mM ammonium formate in water, and the mobile phase B was 0.125% formic acid and 10 mM ammonium formate in 95:5 (v/v) of acetonitrile:water. The column temperature was set to 45 °C, and the gradient elution was conducted as following at a flow rate of 0.4 mL/min: 0 min, 100% of phase B; 0-2 min, 100% of B; 2–7.7 min, 70% of phase B; 7.7–9.5 min, 40% of phase B; 9.5–10.25 min, 30% of phase B; 10.25–12.75 min, 100% of phase B; 12.75–17 min, 100% of phase B; 13 min of post-run. The MS acquisition was operated at targeted-SIM mode using Thermo Scientific Q Exactive Plus Hybrid Quadrupole-Orbitrap mass spectrometer and Xcalibur software (Thermo Scientific Fisher). LPE P-18:0 with hydrogen adduct ([M + H]$^+$) was inputted as the target compound. Detailed instrumental parameters of MS are provided in Supplementary Fig. 23.

**Data processing**. The lipidomics data were processed by MS-DIAL version 4.60[41]. For lipid identification, LipidBlast version 68 was adopted[42]. Both accurate mass similarity score and reverse product score for MS/MS spectra matching were considered to achieve confidence in lipid identification.

LipidBlast can annotate ether phospholipids but cannot identify whether ether phospholipid is plasmanyl form or plasmenyl form (plasmalogen) except plasmanyl PE and plasmenyl PE. To avoid misannotation, plasmenyl LPC and plasmenyl LPE were annotated based on the previous study[43–46]. However, plasmenyl PC cannot be distinguished from plasmanyl PC with MS/MS fragmentation alone. For this reason, we identified only ether phosphatidylcholine peaks disappearing during acid hydrolysis as plasmenyl PC, using the fact that plasmalogen is unstable in the acidic environment. The example of standard fragmentation and lipid annotation are provided in Supplementary Figs. 24 and 25.

For the LPE P-18:0 analysis in mouse serum, the peak area was obtained by MassHunter Qualitative Analysis B.07.00 (Agilent) and normalized by internal standard (LPE 17:1) peak area. Authentic LPE P-18:0 standard was used for calculation of calibration curve generated by datapoint of 0, 1, 5, 10, 50, 100, 500, and 1000 ppb.

For the LPE P-18:0 analysis in cell culture medium, the peak was determined by using authentic LPE P-18:0 standard. The peak area was obtained by El-MAVEN version 0.12.0. The extracellular LPE P-18:0 consumption was calculated as following: (LPE P-18:0 peak area of blank medium – LPE P-18:0 peak area of cell growth medium)/Total cellular protein quantity (mg).

**Statistical analysis of lipidomics data**. The pre-processed data were normalized by the locally weighted scatterplot smoothing (LOWESS) method before inputted into MetaboAnalyst 5.0 for statistical analysis[47]. All detected features with missing values greater than 50% were removed followed by the estimation of remaining features using feature-wise k-Nearest Neighbours (kNN). To raise the quality of data, features with a relative standard deviation higher than 20% or with a S/N ratio below than 10 were filtrated.

Statistical models were created using normalization by median, logarithmic transformation (base 10), and Pareto scaling, and only identified features were used for statistical analysis. Unsupervised principal component analysis (PCA) was applied for data visualization. Unpaired parametric t-test was applied to find significant differences in comparison of groups, and it was determined to be statistically significant only if False Discovery Rate (FDR) <0.05 were satisfied. Volcano plot was created using ggplot2 package (ver. 3.3.3) in R and applied to get lipid peaks satisfying both FDR (<0.05) and Fold change (> 2 or < 0.5). Also, Lipid

ontology[48] enrichment analysis was conducted to find the tendency of lipid subclasses, hence all identified lipids were inputted to enrichment analysis.

**Human samples**. Human intra-abdominal and subcutaneous fat tissues were removed at Surgery Unit of Pusan National University Hospital. Tissues were stored at −20 °C immediately after collection until lipidome analysis. Tissues from 48 individuals were used; population characteristic is available in Supplementary Table 1.

**Statistics**. The number of mice used in animal experiments is described within the associated figure legend. GraphPad Prism 7 software (GraphPad Software, USA) was used for statistical analysis. Data are presented as mean ± standard errors of the mean (SEM) or mean ± standard deviation (SD) as indicated in the Figure Legends. Statistical significance between two groups was determined by unpaired $t$-test. Comparisons among multiple groups were performed using a one-way analysis of variance (ANOVA), with Bonferroni post hoc tests or Tukey's Honestly Significant difference (HSD) post hoc test to determine $p$ values.

**Reporting summary**. Further information on research design is available in the Nature Research Reporting Summary linked to this article.

## Data availability

The RNA-seq data used in this study are available in Gene Expression Omnibus (GEO) database under accession number GSE 182930. Global Transcriptome profiling conducted by Dahlman[49] were analyzed in the study and are publicly available in GEO database under accession number GSE94753. The lipidomics data generated in this study have been deposited in a public database, MetaboLights[50] under accession number MTBLS4703 [ID: yunyochl@snu.ac.kr, PW: NCOMMS-22-00143]. The remaining data are available within the Article, Supplementary information. Source data are provided with this paper.

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

## Acknowledgements

This research was supported by the National Research Foundation of Korea (NRF) grants (NRF-2019R1C1C1002014 to Y.-H.L., NRF-2018R1A5A2024425 to Y.-H.L., NRF-2013M3A9D5072550 to J.K.S., NRF-2012M3A9C4048796 to S.W.K.) funded by the Korean government (MSIT) and National Institutes of Health (NIH) grant (R01DK62292 to J.G.G.).

## Author contributions

Y.-H.L., J.G.G. and S.W.K. conceived and designed the study. Y.K.C., H.I., M.K., Y.S., A.S., C.C. and J.K.S. conducted the animal experiments. S.W.K., Y.P.K., J.H.K., and Y.C.Y. conducted lipidomics analysis. Y.K.C., H.I., M.K., J.L., S.L. and Y.S. performed in vitro experiments. H.K.H., Y.-S.J. and Y.C.Y. analyzed human samples. Y.-H.L., Y.K.C., J.G.G., and Y.C.Y. wrote the manuscript. All authors reviewed the manuscript.

## Competing interests

The authors declare no competing interests.
