## [Peer Review File · Nature Communications]

Title: Adipocyte lysoplasmalogenase TMEM86A regulates plasmalogen homeostasis and protein kinase A-dependent energy metabolismREVIEWER COMMENTS

Reviewer #1 (Remarks to the Author):

20220210 Nature review

The knowledge of adipose biology, essential for understanding the pathophysiological basis of obesity and metabolic diseases, was advanced by giant steps in these highly important studies.

The structure of TMEM86A has close homology to TMEM86B (Wu et al, 2011), a lysoplasmalogenase that catalyzes hydrolysis of lysoplasmalogen to form glycerophospho-choline or -ethanolamine. The function of TMEM86A* (*see note below) and its role in biology are not known.

The authors investigated the regulatory role of this putative lysoplasmalogenase in lipid metabolism using overexpression of the gene (OE) in the C3H10T1/2 adipose cell line and an adipocyte-specific TMEM86A knockout mouse model. Untargeted lipidomic analyses are important in their studies where they quantified plasmenyl-, lysoplasmenyl-, phosphatidyl-, lysophosphatidyl-, plasmanyl-, and lysoplasmanyl-glycerophospholipids in adipose cells and tissues. Their many important and novel findings include:

1. TMEM86A regulates adipose lysoplasmalogen and plasmalogen levels. Overexpression in C3H10T1/2 cells resulted in decreased levels of LPE-P, LPC-P, PE-P and PC-P. When OE cells were challenged with LPE-P, the level of LPE-P decreased providing evidence for lysoplasmalogenase enzymatic activity. * Microscopic analysis showed location of TMEM86A to be in ER (Wu et al., 2011), and mutation of critical amino acids proposed to be involved in TMEM86B catalysis (Jurkowitz et al., 2015) caused decreased rate of lysoplasmalogenase in HEK293 cells.
2. Tmem86A is abundantly expressed in white adipose tissue (WAT) and upregulated by high fat diet (HFD) feeding in mice suggesting the importance of this gene in the pathological response of adipose tissue to overnutrition. Moreover, adult obese women with insulin insensitive diabetes have more mRNA for TMEM86A.
3. Adipose-specific TMEM86A knockout mice have increased lysoplasmalogen and plasmalogen levels in brown and white adipose tissues.
4. In vivo, adipocyte-specific deletion of TMEM86A (KO) mice have increased mitochondrial oxidative metabolism via upregulation of Protein kinase A (PKA) downstream signaling. Evidence included increased phosphorylation of HSL and CREB in BAT (brown adipose tissue) and WAT. Importantly they found in knockout animals increases in whole body energy metabolism, lower percentage of body fat, increased surface temperature in cold conditions, and increased mitochondrial electron transport. Microscopy studies found increased number of mitochondria in KO WAT adipose tissue.
5. In vitro studies on adipocytes isolated from the KO animals confirmed the results in the animal studies: The cells from KO mice have increased phosphorylation of HSL and CREB and increases in mitochondrial proteins involved in electron transport and ATP synthesis.
6. Knockout of TMEM86A protects mice against high fat diet (HFD) obesity partly by increase in PKA signaling in mitochondrial activity. Fed the HFD, KO mice weigh less, their adipose tissue weighs less, glucose tolerance is improved, and they have increased PKA activity and higher levels of phosphorylated CREB and HSL. KO has smaller adipocytes.

7. In vitro treatment of adipocytes with LPE P-18 elevates intracellular cAMP signaling via inhibition of Phosphodiesterase activity (PDE). Treatment results in increase in intracellular cAMP, decrease in PDE, increased expression of mitochondrial protein, and increased oxygen consumption

8. In vivo treatment of mice with LPE P-18 increases mitochondrial oxidative metabolism and protects against diet-induced obesity in mice. Two weeks of intraperitoneal injection of LPE-P-18 raised serum levels of the lysolipid to equal that of KO. The lysolipid caused increase in PKA signaling, increase in phosphorylated proteins (P-HSL and P-CREB) and increases in other mitochondrial proteins involved in mitochondrial energy metabolism in adipose tissue. In humans, they found there is an inverse relationship between BMI Body Mass Index and level of serum LPE-P.

* Concerning the function of TMEM86A: In 2015, a student, Billy Woltz, Chuck Bell, and I cloned and expressed the human TMEM86A gene into E.coli, purified, and characterized the protein. It has abundant lysoplasmalogenase activity with properties nearly identical to those of the liver TMEM86B (Wu et al., 2011) and the bacterium, *L. pneumophila*, (Jurkowitz et al., 2015) lysoplasmalogenase proteins. This work has not been published (Woltz, V., Bell, C.E., Jurkowitz, M. 2015, unpublished work).

Significance of present work:

The present work will be of high significance across many biological fields of study - lipid metabolism, energy metabolism, nutrient metabolism, mitochondria, and pathophysiology of obesity and diabetes and other metabolic diseases, and mitochondrial disease. It is interesting that TMEM86B and TMEM86A, although they are both lysoplasmalogenases, are differentially expressed in the cells and tissues of the body, thus they are not simply redundant genes. They are on different chromosomes too. TMEM86B is highly expressed in the liver, and small intestinal epithelium, and its roles in these tissues and cells are not yet known.

The manuscript is well written, clear and succinct. The references are important and well chosen.

Much evidence is presented that supports the findings. In many instances more than one experimental approach is taken to confirm or support the finding. No more experiments are needed.

I see no flaws in the data analysis, interpretations, and conclusions.

The methodology covers a extremely wide range of procedures and techniques, are well described, and are appropriate for the experiments.

In nearly all cases the methods and procedures are described in enough depth that the reader could repeat the experiment. One exception is line 428, Site directed mutagenesis paragraph. There are not enough details: for example, HEK293 cells are not mentioned.

.....

Overall: please check on the proper use of capital and lower case letters when writing "TMEM86A" and "tmem86a", when to use the one or the other.

Please check your use of "LPC-P", and "LPE-P". In some figures, you are using "LPC" and "LPE" for the lysoplasmeryl lysolipids, and it is confusing. Especially in figure 7, almost every figure 7A through 7J the terms "LPC or LPE" are used for "LPC-P and LPE-P" and it is very confusing, even though you explain in the Figure 7 legend.

Please check on your labels of the Y axes in your figures. Check that you describe them in more detail where needed or possible. For example: Figure 5E, the pmoles are what chemical? and in Figure 8A, what are "ppb"? This figure could also include "serum" as a heading. It's nice to have the information on the figures, rather than having to search manuscript for explanations. In Figure 2B-D, what does the Y axis label, "Ppia" stand for?

Below I will suggest some changes or describe minor problems

Lines:

54. "plasmalogens" aren't "further matured". The "intermediate" is further matured in the ER, becoming plasmalogen in the ER.

72. Describe the C3H10T1/2 cell type here.

103. ...activity in HEK293 cells (Figure 1J-K). Add HEK293 cells.

149. ...plasmanyl PE....this is not in figure.

233. ...add "M1"

234. add "M2"

283. spell out "body mass index" "(BMI)"

299. they are not "histidine" residues, rather "aspartate" for "D"

I think your work is exceptionally fine research! I recommend it be published in Nature Communications.

Marianne Shaw Jurkowitz

Reviewer #2 (Remarks to the Author):

In the manuscript entitled "Adipocyte-specific deletion of lysoplasmalogenase TMEM86A protects mice against obesity-induced metabolic dysfunction", Cho, Yoon and Im, et al, aimed to investigate whether plasmalogens metabolism interferes in adipose tissue function and impact obesity and type 2 diabetes progress.

Overall, I believe the authors were able to put together an interesting study. Although the manuscript describes quite well the entire phenomenon, there are some important missing points that must be clarified and revised according to the comments that follows below.

- I believe the first change needed is in the title. The title chosen is actually undermining the whole story. The manuscript suggests a physiological role for TMEM86A and for LPE P18:0 in the regulation of adenylate cyclase-cAMP-PKA axis in adipocytes. The findings demonstrate more than a simple consequence of the enzyme deletion for obesity prevention, as stated in the title.

- Introduction is quite brief and must be improved. I don't think the importance of this investigation is really well justified in the introduction. In addition, a clear hypothesis is also missed. In the very brief description of the study in the last sentence of the introduction, the authors limited the description to the tissue-specific gene deletion and lipidomics. As in the title, it is undermining the actual findings.

- The plasmalogen effects in thermogenesis was already described by Park et al., 2019 (JCI), and the effect of lysoplasmalogen as PKA activators were already known (Williams and Ford, 1997, FEBS Letters). That means the current results were somehow predictable. These studies may need to be properly acknowledged in the introduction to position the present manuscript in the actual state of the art. Since most of the effects here observed were in the white fat, I believe the novelty is not really harmed by those previous studies, but those must be recognized in the introduction.

- Seems that the Authors made a wrong choice in adding the TMEM86A expression data in the figure 2. That is a more introductory one, and would makes more sense if those data comes first than the in vitro overexpression data (current fig 1). This would definitely better introduce the study and justify why the Authors decided to overexpress and further Knockout this gene.

- In Figure 4, what were the effects of the gene deletion in total body weight? This information is also important for the interpretation of the energy balance data, since those are normalized by kg. If lean mass and activity data were obtained, the energy expenditure data can be normalized by both. Interestingly, the traces show that EE is higher only in light cycles (Fig 4G). To better understand it, might be needed to provide the traces for the activity too.

- It is unexpected to see higher RER in animal models that displays elevated thermogenesis and upregulated sympathetic pathways in BAT and WAT. As can be seen after CL injection (Fig 4H), cAMP-PKA-dependent thermogenic stimuli turn the metabolic preference towards the fatty acid oxidation (lower RER), rather than carbohydrate oxidation. Considering the Authors repeatedly demonstrated the AKO or LPE 18:0 increases cAMP-dependent pathways, how could the RER data be explained?

- Acute effect of CL on VO₂, VCO₂ and EE could be better demonstrated by an AUC quantification or delta of increase in comparison to the baseline immediately before injection.

- Data on liver weight and histology under high-fat diet is a really important piece of data missed in this study.

- In Fig 5, Seahorse analysis could be improved by the quantification of proton leak (uncoupling) which is

clearly changed.

- It is a pity that the authors did not assess insulin sensitivity and fasting glucose in this model. The activation of the sympathetic downstream pathways and reduction of inflammation in WAT, as here described, usually result in metabolic phenotype accompanied by improved insulin sensitivity and lower glucose levels. This is another data that should be provided, even if negative.

- Fig 6: Providing the M2/M1 macrophage ratio would definitely improve the way your data is demonstrated.

- In the last figure, more detailed phenotyping is missing. It is surprising that after treating mice for days with the lipid, the authors did not measure the effects in body weight, adipose tissue depots weights, energy balance and insulin sensitivity. It is ok if it does not phenocopy exactly the AKO mice, since there might have other lipids influencing the AKO metabolic phenotype, but one cannot resign to show such a data.

- The text must also be revised in terms of grammar. It is totally understandable such kind of issues since the authors are not native speakers, but a proofread is needed.

Reviewer #3 (Remarks to the Author):

In this study, Cho et al used lipidomics analysis, adipose tissue-specific knockout mice, and deletion and overexpression cells to investigate the role of TMEM86A in metabolism. They identified TMEM86A as a major lysoplasmalogenase in adipose tissue and adipocyte-specific deletion of TMEM86A protects mice against high fat diet-induced metabolic dysfunction. Furthermore, they demonstrated that knockout of Tmem86a or treated mice with plasmeyl lysophosphatidylethanolamine 18:0 (LPE P-18:0), a major product of TMEM86A in mouse adipose tissue, enhanced the PKA signaling pathway by inhibiting PDE, leading to elevated thermogenesis and energy expenditure. Lastly, they found that LPE P-18:0 levels were significantly lower in adipose tissue of human obese subjects, suggesting that TMEM86A could be a target for the prevention and treatment of obesity-related metabolic diseases. In general, these experiments were well executed, and the experimental data are largely supportive of the conclusion. This work could be further improved if the following questions could be addressed.

1. What is the tissue distribution of TMEM86A? Whether HFD feeding increases TEME86A expression in other tissues in addition to adipose tissue?
2. While Tmem86a mRNA was heavily enriched in adipocytes, lower levels of the mRNA could also be detected in SVFs (Figure 2I and J). Whether HFD feeding has any effect on Tmem86a mRNA and protein expression in SVFs?
3. TMEM86A KO led to a significantly upregulation of lysoplasmalogens or plasmalogens in adipose

tissue. Are there differences in adipose lysoplasmalogen or plasmalogen levels between HFD-fed mice and normal control mice?

4. The authors found that the expression levels of TME86A are very low in BAT and HFD feeding had no effect on Tmem86a expression in BAT (Fig. 2B and 2E). However, TMEM86A KO significantly increased phosphorylation of HSL and CREB in BAT (Figure 4A-D). In addition, Tmem86a KO increased Ucp1 expression in BAT but not gWAT (Fig. 4A-C), but the effects of Tmem86a KO on mitochondrial protein expression were most robust in primary white adipocytes derived from gWAT (Figure 5A-D). These results are somewhat confusing and should be clarified.

5. Treatment of C3H10T1/2 adipocytes with LPC P-18:0 or LPE P-18:0 significantly increased levels of phosphorylated HSL, CREB, and PKA substrates (Figure 7A-B). Could TMEM86A overexpression increase LPE P-18:0 levels in adipocytes? Is LPE P-18:0 able to stimulate the phosphorylation of HSL, CREB, and PKA substrates and expression levels of Ucp-1 in Tmem86a deficient cells?

6. The authors found that LPE P-18:0 inhibits PDE. Which PDE isoform is inhibited by LPE P-18:0?

7. The authors found that overexpression of TMEM86A in C3H10T1/2 reduced phosphorylation of HSL, CREB, and PKA substrates. Does overexpression of TME86A activate PDE and reduce cAMP levels?

We thank the Editors and Reviewers for their valuable comments and the opportunity to improve our manuscript. We have undertaken extensive revision, including 8 new supplementary figures that address concerns raised in the review. We also revised the Title, Abstract, and Introduction.

The major revisions include the followings:

- qPCR analysis of tissue distribution of *Tmem86a* expression in vivo (Supplementary Figure 1)
- Detailed indirect calorimetry analysis of TMEM86A KO and wild type (WT) mice (Supplementary Figure 12), including new indirect calorimetry analyses under ad libitum feeding and 24-hour fasting conditions, provided in the point-by-point responses (Figure R3-4)
- Intraperitoneal insulin tolerance test, immunoblot analysis of Insulin sensitivity, and hepatic phenotype analysis of TMEM86A KO and WT mice fed a HFD for 8 weeks (Supplementary Figure 14)
- cAMP levels in TMEM86A-overexpressing adipocytes (Supplementary Figure 16)
- qPCR and immunoblot analyses of PDE3B expression levels in adipose tissue (Supplementary Figure 18)
- Immunoblot analysis of effects of cilostamide treatment on LPE P-18:0-induced phosphorylation of PKA downstream substrates (Supplementary Figure 19)
- Untargeted lipidomics analysis of phospholipids in BAT, iWAT, and gWAT of mice fed a NCD or HFD for 8 weeks (Figure R9-12). And targeted lipidomics analysis of LPE P-18:0 in BAT, iWAT, and gWAT of mice fed a NCD or HFD for 8 weeks (Supplementary Figure 20)
- Intraperitoneal insulin tolerance test and indirect calorimetry analysis of HFD-fed mice, treated with LPE P-18:0 (Supplementary Figure 21)

Source data are provided with the revised manuscript (Source data for 8 main figures and supplementary figures).

All mass spectrometry raw data of lipidomics profiling were deposited to Metabolomics and the accession number is provided in this manuscript.

Reviewer #1 (Remarks to the Author):

The knowledge of adipose biology, essential for understanding the pathophysiological basis of obesity and metabolic diseases, was advanced by giant steps in these highly important studies.

The structure of TMEM86A has close homology to TMEM86B (Wu et al, 2011), a lysoplasmalogenase that catalyzes hydrolysis of lysoplasmalogen to form glycerophosphocholine or -ethanolamine. The function of TMEM86A* (*see note below) and its role in biology are not known.

The authors investigated the regulatory role of this putative lysoplasmalogenase in lipid metabolism using overexpression of the gene (OE) in the C3H10T1/2 adipose cell line and an adipocyte-specific TMEM86A knockout mouse model. Untargeted lipidomic analyses are important in their studies where they quantified plasmenyl-, lysoplasmenyl-, phosphatidyl-, lysophosphatidyl-, plasmanyl-, and lysoplasmanyl-glycerophospholipids in adipose cells and tissues. Their many important and novel findings include:

1. TMEM86A regulates adipose lysoplasmalogen and plasmalogen levels. Overexpression in C3H10T1/2 cells resulted in decreased levels of LPE-P, LPC-P, PE-P and PC-P. When OE cells were challenged with LPE-P, the level of LPE-P decreased providing evidence for lysoplasmalogenase enzymatic activity. *

Microscopic analysis showed location of TMEM86A to be in ER (Wu et al., 2011), and mutation of critical amino acids proposed to be involved in TMEM86B catalysis (Jurkowitz et al., 2015) caused decreased rate of lysoplasmalogenase in HEK293 cells.

2. Tmem86A is abundantly expressed in white adipose tissue (WAT) and upregulated by high fat diet (HFD) feeding in mice suggesting the importance of this gene in the pathological response of adipose tissue to overnutrition. Moreover, adult obese women with insulin insensitive diabetes have more mRNA for TMEM86A.

3. Adipose-specific TMEM86A knockout mice have increased lysoplasmalogen and plasmalogen levels in brown and white adipose tissues.

4. In vivo, adipocyte-specific deletion of TMEM86A (KO) mice have increased mitochondrial oxidative metabolism via upregulation of Protein kinase A (PKA) downstream signaling. Evidence included increased phosphorylation of HSL and CREB in BAT (brown adipose tissue) and WAT. Importantly they found in knockout animals increases in whole body energy metabolism, lower percentage of body fat, increased surface temperature in cold conditions, and increased mitochondrial electron transport. Microscopy studies found increased number of mitochondria in KO WAT adipose tissue.

5. In vitro studies on adipocytes isolated from the KO animals confirmed the results in the animal studies: The cells from KO mice have increased phosphorylation of HSL and CREB and increases in mitochondrial proteins involved in electron transport and ATP synthesis.

6. Knockout of TMEM86A protects mice against high fat diet (HFD) obesity partly by increase in PKA signaling in mitochondrial activity. Fed the HFD, KO mice weigh less, their adipose tissue weighs less, glucose tolerance is improved, and they have increased PKA activity and higher levels of phosphorylated CREB and HSL. KO has smaller adipocytes.

7. In vitro treatment of adipocytes with LPE P-18 elevates intracellular cAMP signaling via inhibition of Phosphodiesterase activity (PDE). Treatment results in increase in intracellular cAMP, decrease in PDE, increased expression of mitochondrial protein, and increased oxygen consumption

8. In vivo treatment of mice with LPE P-18 increases mitochondrial oxidative metabolism and protects against diet-induced obesity in mice. Two weeks of intraperitoneal injection of LPE-P-18 raised serum levels of the lysolipid to equal that of KO. The lysolipid caused increase in PKA signaling, increase in phosphorylated proteins (P-HSL and P-CREB) and increases in other

mitochondrial proteins involved in mitochondrial energy metabolism in adipose tissue. In humans, they found there is an inverse relationship between BMI Body Mass Index and level of serum LPE-P.

* Concerning the function of TMEM86A: In 2015, a student, Billy Woltz, Chuck Bell, and I cloned and expressed the human TMEM86A gene into E.coli, purified, and characterized the protein. It has abundant lysoplasmalogenase activity with properties nearly identical to those of the liver TMEM86B (Wu et al., 2011) and the bacterium, L. pneumophila, (Jurkowitz et al., 2015) lysoplasmalogenase proteins. This work has not been published (Woltz, V., Bell, C.E., Jurkowitz, M. 2015, unpublished work).

Significance of present work:

The present work will be of high significance across many biological fields of study - lipid metabolism, energy metabolism, nutrient metabolism, mitochondria, and pathophysiology of obesity and diabetes and other metabolic diseases, and mitochondrial disease. It is interesting that TMEM86B and TMEM86A, although they are both lysoplasmalogenases, are differentially expressed in the cells and tissues of the body, thus they are not simply redundant genes. They are on different chromosomes too. TMEM86B is highly expressed in the liver, and small intestinal epithelium, and its roles in these tissues and cells are not yet known.

The manuscript is well written, clear and succinct. The references are important and well chosen.

Much evidence is presented that supports the findings. In many instances more than one experimental approach is taken to confirm or support the finding. No more experiments are needed.

I see no flaws in the data analysis, interpretations, and conclusions.

The methodology covers a extremely wide range of procedures and techniques, are well described, and are appropriate for the experiments. In nearly all cases the methods and procedures are described in enough depth that the reader could repeat the experiment.

Response: We thank the reviewer for the insightful and encouraging comments on our manuscript. In accordance with the reviewer's suggestion, we revised the manuscript as detailed below.

Q1. One exception is line 428, Site directed mutagenesis paragraph. There are not enough details: for example, HEK293 cells are not mentioned.

Response 1: *We included a more detailed description of the methods “site-directed mutagenesis”. (page 22)*

Q2. Overall: please check on the proper use of capital and lower case letters when writing "TMEM86A" and "tmem86a", when to use the one or the other.

Response 2: *According to the reviewer’s comment, we corrected the inconsistent use of capital and lower-case letters. In this revised manuscript, we corrected the inconsistency, following the guidelines for the symbols of protein and gene (references: NOMENCLATURE. Genomics 45, 471-476 (1997)(<https://doi.org/10.1006/geno.1997.5010>); International Protein Nomenclature Guidelines (https://www.ncbi.nlm.nih.gov/genome/doc/internatprot_nomenguide/); Maltais, L.J., Blake, J.A., Eppig, J.T. & Davisson, M.T.)*

*For example, we used “**TMEM86A**” for proteins (for vertebrates, use an all uppercase gene symbol in a protein name, and it is not italicized).*

*For the mRNA transcript of mice, we used “**Tmem86a**” (Mouse gene symbols are italicized with only the first letter in upper-case).*

*For the mRNA transcript of humans, we used “**TMEM86A**” (Human gene symbols are italicized, with all letters in uppercase).*

Q3. Please check your use of "LPC-P", and "LPE-P". In some figures, you are using "LPC" and "LPE" for the lysoplasmeryl lysolipids, and it is confusing. Especially in figure 7, almost every figure 7A through 7J the terms "LPC or LPE" are used for "LPC-P and LPE-P" and it is very confusing, even though you explain in the Figure 7 legend.

Response 3: *According to the reviewer’s comment, we replaced “LPC” and “LPE” with “LPC-P” and “LPE-P” in Figure 7.*

Revised Fig. 7

Q4. Please check on your labels of the Y axes in your figures. Check that you describe them in more detail where needed or possible. For example: Figure 5E, the pmoles are what chemical? and in Figure 8A, what are "ppb"? This figure could also include "serum" as a heading. It's nice to have the information on the figures, rather than having to search manuscript for explanations. In Figure 2B-D, what does the Y axis label, "Ppia" stand for?

Response 4: According to the reviewer's comment, we add the information as shown below

We added OCR for y-axis (Figure 5e, 5j, and 7i), replaced ppb with "concentration (ug/L)" (Figure 8a), and included the full name for Ppia (Peptidylprolyl isomerase A) in the method section (page 27) and figure legends (revised Figure 1)

Below I will suggest some changes or describe minor problems

Lines:

Q5. Line 54. "plasmalogens" aren't "further matured". The "intermediate" is further matured in the ER, becoming plasmalogen in the ER.

Response 5: According to the reviewer's comment, we corrected "plasmalogens" into "the intermediate products".

Q6. Line 72. Describe the C3H10T1/2 cell type here.

Response 6: According to the reviewer's comment, we added the description of the C3H10T1/2 cell type. (page 7): "In this analysis, we used adipocytes differentiated from C3H10T1/2 cells, a cloned murine embryo fibroblast cell line"

Q7. Line 103. ...activity in HEK293 cells (Figure 1J-K). Add HEK293 cells.

Response 7: According to the reviewer's comment, we added "HEK293T cells". (page 8)

Q8. Line 149. ...plasmany PE....this is not in figure.

Response 8: We apologize for the wrong information. We corrected "Plasmany PE" into "Plasmany LPC", as indicated below, and cited Supplementary Fig. 9.

Q9. Line 233. ...add "M1", 234. add "M2"

Response 9: According to the reviewer's comment, we added "M1" and "M2".

Q10. Line 283. spell out "body mass index" "(BMI)"

Response 10: According to the reviewer's comment, we spell out "body mass index".

Q11. Line 299. they are not "histidine" residues, rather "aspartate" for "D"

Response 11: We apologize for the wrong information. We corrected "histidine" into "aspartate"

I think your work is exceptionally fine research! I recommend it be published in Nature Communications. Marianne Shaw Jurkowitz

Response: We really appreciated your elegant research work that has pioneered the field of plasmalogen metabolism including for the first time identification of lysoplasmalogenases. We also appreciated that your publications have been a gold standard and a great foundation for the current work.

Reviewer #2 (Remarks to the Author):

In the manuscript entitled “Adipocyte-specific deletion of lysoplasmalogenase TMEM86A protects mice against obesity-induced metabolic dysfunction”, Cho, Yoon and Im, et al, aimed to investigate whether plasmalogen metabolism interferes in adipose tissue function and impact obesity and type 2 diabetes progress.

Overall, I believe the authors were able to put together an interesting study. Although the manuscript describes quite well the entire phenomenon, there are some important missing points that must be clarified and revised according to the comments that follows below.

Q1. I believe the first change needed is in the title. The title chosen is actually undermining the whole story. The manuscript suggests a physiological role for TMEM86A and for LPE P18:0 in the regulation of adenylate cyclase-cAMP-PKA axis in adipocytes. The findings demonstrate more than a simple consequence of the enzyme deletion for obesity prevention, as stated in the title.

Response 1: We thank the reviewer for the constructive comments. We revised our title to be concise and informative.

New Title: Adipocyte lysoplasmalogenase TMEM86A regulates plasmalogen homeostasis and PKA-dependent energy metabolism.

Q2. Introduction is quite brief and must be improved. I don't think the importance of this investigation is really well justified in the introduction. In addition, a clear hypothesis is also missed. In the very brief description of the study in the last sentence of the introduction, the authors limited the description to the tissue-specific gene deletion and lipidomics. As in the title, it is undermining the actual findings.

Response 2: We revised the introduction according to the reviewer's comment. (Page 4-5)

Q3. The plasmalogen effects in thermogenesis was already described by Park et al., 2019 (JCI), and the effect of lysoplasmalogen as PKA activators were already known (Williams and Ford, 1997, FEBS Letters). That means the current results were somehow predictable. These studies may need to be properly acknowledged in the introduction to position the present manuscript in the actual state of the art. Since most of the effects here observed were in the white fat, I believe the novelty is not really harmed by those previous studies, but those must be recognized in the introduction.

Response 3: According to the reviewer's suggestion, we acknowledged the works by Park et al. (2019 JCI) and Williams and Ford (1997, FEBS Letters) in the introduction of the revised manuscript (page 5).

Our original manuscript cited Williams and Ford (1997) in the discussion, as it first suggested a possible regulation of PKA by lysoplasmenylcholine, and such a mechanism was consistent with the elevation of PKA signaling we observed. However, we demonstrate that the mechanism

proposed by Williams and Ford is unrelated to activation of PKA signaling by TMEM86a knockout. Williams and Ford investigated the direct activation of purified PKA holoenzyme, and found in that LPC P-, but not LPE P-, activated the purified complex. In contrast, our work demonstrates that both LPE P- and LPC P- activates PKA by increasing cAMP levels. Furthermore, LPE P- inhibited PDE activity (PDE3B activity assay in cell-free system). We hope that the reviewer will agree that the work of Williams and Ford is mechanistically irrelevant to the current study and that our results are by no means predictable by from that work.

Thus, we revised our manuscript to emphasize that the previous study characterized direct and cAMP-independent effects of LPC P- on PKA activation, which was not in the context of actual PKA-dependent signaling pathways.

Introduction: "Previously, Williams and Ford investigated the direct activation of purified PKA holoenzyme, and found that lysoplasmenylcholine activated the purified complex. However, whether and how lysoplasmalogens might influence PKA signaling in adipocytes has not been investigated."

Discussion: "Previous studies indicated that LPC-P directly increases PKA activity, which was shown by assay with purified PKA proteins in cell-free system¹². In this study, LPE-P or other lysophospholipid classes do not activate PKA activity, indicating specificity of PKA for lysophospholipids having choline polar head groups. However, the effects of lysoplasmalogens on PKA-dependent lipid metabolism in the context of cellular signaling were not determined."

Q4. Seems that the Authors made a wrong choice in adding the TMEM86A expression data in the figure 2. That is a more introductory one, and would makes more sense if those data comes first than the in vitro overexpression data (current fig 1). This would definitively better introduce the study and justify why the Authors decided to overexpress and further Knockout this gene.

Response 4: *We thank the reviewer for the suggestion to give a better logical flow to the manuscript. According to the reviewer's suggestion, we rearranged the figures in a new order (Figure 1 ⇔ Figure 2)*

Q5. In Figure 4, what were the effects of the gene deletion in total body weight? This information is also important for the interpretation of the energy balance data, since those are normalized by kg. If lean mass and activity data were obtained, the energy expenditure data can be normalized by both. Interestingly, the traces show that EE is higher only in light cycles (Fig 4G). To better understand it, might be needed to provide the traces for the activity too.

Response 5: *According to the comments, we provide a bar graph that indicates body weight of WT and KO mice that underwent indirect calorimetry. The average body weight of TMEM86A KO mice was significantly lower than WT mice (**Supplementary Fig. 12a**).*

In this revision, we provided the time course of the activity in Supplementary Fig. 12c. The activity was higher in WT during the night time (12-18 hr) (Fig. R1) and after CL treatment (Fig. R1), thus difference in activity cannot account for the elevated VO₂ in KO. According to the reviewer's suggestion, data were presented as energy expenditure normalized by lean mass and activity, shown below (Fig. R2).

Q6. It is unexpected to see higher RER in animal models that displays elevated thermogenesis and upregulated sympathetic pathways in BAT and WAT. As can be seen after CL injection (Fig 4H), cAMP-PKA-dependent thermogenic stimuli turn the metabolic preference towards the fatty acid oxidation (lower RER), rather than carbohydrate oxidation. Considering the Authors repeatedly demonstrated the AKO or LPE 18:0 increases cAMP-dependent pathways, how could the RER data be explained?

Response 6: We appreciate the reviewer's comments regarding RER. RER is influenced by many interrelated factors, including diet, time of day relative to food intake, level of adrenergic activation, and physical activity among others. In view of the reviewer's comments, we performed detailed analysis of several possible factors to better understand daily variation on RER of WT and KO mice. In **Fig. R3**, we noticed that higher levels of RER of KO mice (0-6, 6-12, and 42-48 hours) are associated with higher food intake of KO mice. RER values of WT and KO mice between 12 to 18 hours were not significantly different as KO mice showed lower food intake and activity. Importantly, over a 24-hour period, we found no difference in RER, which is expected since daily RER reflects the balance between lipid synthesis and oxidation, which ultimately must match the diet composition.

We performed additional indirect calorimetry analyses (under ad libitum feeding and fasting conditions) to further confirm our data. As shown below, within the 24-hour period of ad libitum feeding, we observed the expected variations in RER that reflect the fed and fasted states (**Fig. R4**). We note, however, that the decline in RER seen in the lights-on period (reflecting sleep/fasting/inactivity) was delayed in the KO mice (**Fig. R4**), and this corresponded to the continuation of feeding in the light period (**Fig. R4**). Furthermore, CL tests were performed at the nadir of RER during the lights-on phase; thus, RER was not reduced further since these animals were already in a fasted state. As expected, CL induced a greater increase in VO₂ in KO compared to WT mice.

In contrast, fasting resulted in similar levels of RER between WT and KO over the time course (Figure R5). VO₂, VCO₂, and energy expenditure were significantly higher in KO mice, which is consistent with indirect calorimetry data of feeding conditions mentioned above.

Q7. The acute effect of CL on VO₂, VCO₂, and EE could be better demonstrated by an AUC quantification or delta of increase compared to the baseline immediately before injection.

Response 7: According to the reviewer's comments, we provide AUC quantification of VO₂, VCO₂, and EE. AUC was quantified from the baseline to 4 hours of CL injection (**Supplementary Fig. 12b**). Acute CL effect was significantly higher in TMEM86A AKO mice compared to WT mice.

Q8. Data on liver weight and histology under a high-fat diet is a critical piece of data missed in this study.

Response 8: As requested by the reviewer, we provide bar graphs of liver weight and micrographic images of H&E-stained liver paraffin sections from HFD-fed WT and TMEM86A AKO mice (**Supplementary Fig. 14a, d and e**). TMEM86A AKO reduced HFD-induced increase of liver weight and TG accumulation in liver.

Supplementary Fig. 14. Adipocyte-specific TMEM86A KO protects mice against HFD-induced metabolic dysfunction (related to Fig. 6)

a Liver Weight. n = 6 **d** Representative images of H&E stained paraffin sections of liver from WT and TMEM86A AKO mice, along with magnified view of the boxed regions. Scale bars = 50 μ m. **e** Triglyceride levels in liver from WT and TMEM86A AKO mice. n = 4

Q9. In Fig 5, Seahorse analysis could be improved by the quantification of proton leak (uncoupling) which is clearly changed.

Response 9: In response to the reviewer's comments, we provide bar graphs of proton leak-related OCR (Fig. 5f, 5k, and 7j)

Q10. It is a pity that the authors did not assess insulin sensitivity and fasting glucose in this model. The activation of the sympathetic downstream pathways and reduction of inflammation in WAT, as here described, usually result in metabolic phenotype accompanied by improved insulin sensitivity and lower glucose levels. This is another data that should be provided, even if negative.

Response 10: As pointed out by the reviewer, we assumed that TMEM86A AKO improved insulin sensitivity of HFD-fed mice, but did not confirm it by experiments. We have now addressed this possibility experimentally and found that TMEM86A AKO improved insulin sensitivity as demonstrated by insulin tolerance test (ITT) and insulin signaling via immunoblotting (Supplementary Fig. 14b-c).

Fasting glucose levels were originally provided as a part of GTT analysis (Fig. 6c), and we provide the data separately as a bar graph of fasting glucose levels, shown below (Fig. R6).

Q11. Fig 6: Providing the M2/M1 macrophage ratio would definitely improve the way your data is demonstrated.

Response 11: As suggested by the reviewer, we provide M2/M1 macrophage ratio as Fig. 6l.

Q12. In the last figure, more detailed phenotyping is missing. It is surprising that after treating mice for days with the lipid, the authors did not measure the effects in body weight, adipose tissue depots weights, energy balance and insulin sensitivity. It is ok if it does not phenocopy exactly the AKO mice, since there might have other lipids influencing the AKO metabolic phenotype, but one cannot resign to show such a data.

Response 12: According to the reviewer's comments, we provide data of BW and fat pad weight, food intake, ITT, and indirect calorimetry analysis of high fat-fed mice treated with LPE P-18:0, and provided as Supplementary Fig. 21.

Q13. The text must also be revised in terms of grammar. It is totally understandable such kind of issues since the authors are **not native speakers**, but a proofread is needed.

Response 13: *The manuscript was carefully reviewed by English native speakers.*

Reviewer #3 (Remarks to the Author):

In this study, Cho et al used lipidomics analysis, adipose tissue-specific knockout mice, and deletion and overexpression cells to investigate the role of TMEM86A in metabolism. They identified TMEM86A as a major lysoplasmalogenase in adipose tissue and adipocyte-specific deletion of TMEM86A protects mice against high fat diet-induced metabolic dysfunction. Furthermore, they demonstrated that knockout of *Tmem86a* or treated mice with plasmenyl lysophosphatidylethanolamine 18:0 (LPE P-18:0), a major product of TMEM86A in mouse adipose tissue, enhanced the PKA signaling pathway by inhibiting PDE, leading to elevated thermogenesis and energy expenditure. Lastly, they found that LPE P-18:0 levels were significantly lower in adipose tissue of human obese subjects, suggesting that TMEM86A could be a target for the prevention and treatment of obesity-related metabolic diseases. In general, these experiments were well executed, and the experimental data are largely supportive of the conclusion. This work could be further improved if the following questions could be addressed.

Q1. What is the tissue distribution of TMEM86A? Whether HFD feeding increases TME86A expression in other tissues in addition to adipose tissue?

Response 1: *We thank the reviewer for the constructive comments.*

We examined the effects of HFD feeding on TMEM86A expression in other tissues (liver, kidney, heart, and spleen) and found slight but statistically significant increases in TMEM86A expression levels in the liver and heart. The magnitude of the diet effect is far greater in WAT (Supplementary Fig.1)

Q2. While *Tmem86a* mRNA was heavily enriched in adipocytes, lower levels of the mRNA could also be detected in SVFs (Figure 2I and J). Whether HFD feeding has any effect on *Tmem86a* mRNA and protein expression in SVFs?

Response 2: We did not observe any significant changes in *Tmem86a* expression levels in stromovascular fractions obtained from BAT, iWAT and gWAT of mice fed a NCD or HFD for 8 weeks (Fig. R7). In addition, publicly available global transcriptomic data indicated that there was no difference in *TMEM86A* expression levels in macrophage-(F480+) fractions of gWAT between NCD- and HFD-fed mice (Fig. R8)

In the current study, we focused on the effects of adipocyte-specific *TMEM6A* KO by using an *Adipoq-Cre* mediated KO model.

Q3. *TMEM86A* KO led to a significantly upregulation of lysoplasmalogens or plasmalogens in adipose tissue. Are there differences in adipose lysoplasmalogen or plasmalogen levels between HFD-fed mice and normal control mice?

Response 3. We performed untargeted lipidomics analysis of phospholipids in BAT, iWAT and gWAT of mice fed a NCD or HFD for 8 weeks. Data indicated that HFD feeding reduced several species of plasmalogens and lysoplasmalogens, including LPE P-18:0 (Fig. R9-12). The magnitude of the HFD effect on LPE P-18:0 is greater in WAT.

To further confirm the effects of HFD on LPE P-18:0 levels, we also measured LPE-P 18:0 levels in BAT, iWAT, and gWAT of mice fed a NCD or HFD for 8 weeks, using targeted lipidomics analysis, and found that LPE-P 18:0 levels were significantly lower in iWAT and gWAT of HFD-fed (Supplementary Fig. 20).

Q4. The authors found that the expression levels of TME86A are very low in BAT and HFD feeding had no effect on Tmem86a expression in BAT (Fig. 2B and 2E). However, TMEM86A KO significantly increased phosphorylation of HSL and CREB in BAT (Figure 4A-D). In addition, Tmem86a KO increased Ucp1 expression in BAT but not gWAT (Fig. 4A-C), but the effects of Tmem86a KO on mitochondrial protein expression were most robust in primary white adipocytes derived from gWAT (Figure 5A-D). These results are somewhat confusing and should be clarified.

Response 4: We thank the reviewer for the constructive comment.

8 weeks of HFD feeding did not affect expression levels of TMEM86A in BAT. This result suggested that HFD feeding might induce an upstream regulator of TMEM86A expression, and these effects were specific to WAT, not BAT

Basal mitochondrial content in gWAT of mice is lowest among three adipose tissue depots examined (Ref: Lee Y-H, Mottillo EP, Granneman JG. Adipose tissue plasticity from WAT to BAT and in between. *Biochimica et Biophysica Acta (BBA) - Molecular Basis of Disease*. 2014;1842:358-69; Forner F et.al, Proteome Differences between Brown and White Fat Mitochondria Reveal Specialized Metabolic Functions. *Cell Metabolism*. 2009;10:324-35.) Thus, the magnitude of the increase is inversely proportional to the basal level, as expected.

UCP1 is a brown adipocyte specific marker and UCP1-induction by thermogenic stimuli defines beige adipocytes. (REFs. Cannon B, Nedergaard J. Brown adipose tissue: function and physiological significance. *Physiological reviews*. 2004;84:277-359; Shabalina Irina G, Nedergaard J, et.al., UCP1 in Brite/Beige Adipose Tissue Mitochondria Is Functionally Thermogenic. *Cell Reports*. 2013;5:1196-203.)

For example, UCP1 expression is induced by cold exposure in subcutaneous iWAT while gWAT is resistant to browning/beiging with no detectable UCP1 expression upon cold exposure. UCP1

induction in gWAT requires intense pharmacological stimulation and is not usually observed under physiological conditions.

Q5. Treatment of C3H10T1/2 adipocytes with LPC P-18:0 or LPE P-18:0 significantly increased levels of phosphorylated HSL, CREB, and PKA substrates (Figure 7A-B). Could TMEM86A overexpression increase LPE P-18:0 levels in adipocytes? LPE P-18:0 able to stimulate the phosphorylation of HSL, CREB, and PKA substrates and expression levels of Ucp-1 in Tmem86a deficient cells?

Response 5: *In our original manuscript, we provided untargeted lipidomics data of TMEM86A overexpressing C3H10T1/2 adipocytes (Fig. 1a: revised Fig. 2a), showing reduction in LPE P-18:0 levels by TMEM86A expression.*

According to the reviewer's comment, we performed new experiments to determine the effects of LPE-P 18:0 on PKA signaling in TMEM86A deficient cells (Fig. R13). LPE-P 18:0 treatment did not further increase PKA signaling in TMEM86A KO adipocytes. Both TMEM86A KO and LPE P-18:0 increased PKA signaling through the same mechanism of PDE inhibition, thus there were no additive effects of TMEM86A KO on LPE P-18:0-induced phosphorylation levels of PKA substrates, including p-CREB, p-HSL. LPE P-18:0 treatment did not induce UCP1 expression in in vitro cultures of white adipocytes, while we detected upregulation of UCP1 expression in BAT and iWAT by LPE P-18:0 in vivo (Fig. 4b). We speculated that the intact in vivo tissue environment might be required for UCP1 induction in white adipocytes.

Q6. The authors found that LPE P-18:0 inhibits PDE. Which PDE isoform is inhibited by LPE P-18:0?

Response 6: We apologize that we did not clearly indicate the isotype of PDE used in the experiment. In this original manuscript, we assayed the inhibitory effect of LPE P-18:0 on PDE3B in cell free assay system. We selected to use PDE3B because the major PDE in adipose tissue is PDE3B (highest abundance among PDE isotypes (**Supplementary Fig. 18**)).

In this revised manuscript, we performed new experiments using PDE3B specific inhibitor cilostamide to test the effect of PDE3B inhibition on LPE-P induced PKA signaling pathway activation (**Supplementary Fig. 19**). Cilostamide treatment did not show any additive effects on LPE P-18:0-induced upregulation of p-HSL and p-CREB, suggesting that these two molecules act by the same mechanism (by inhibition of PDE3B).

Q7. The authors found that overexpression of TMEM86A in C3H10T1/2 reduced phosphorylation of HSL, CREB, and PKA substrates. Does overexpression of TEME86A activate PDE and reduce cAMP levels?

Response 7: *In this revision, we performed new experiment to monitor cAMP levels in C3H10T1/2 cells and demonstrated a reduction in cAMP levels by TMEM86A overexpression (Supplementary Fig. 16).*

REVIEWERS' COMMENTS

Reviewer #1 (Remarks to the Author):

20220517

Review of resubmission.

Dear Authors,

You have addressed all of my concerns or changes I suggested in the first submission. This research work is highly important contribution to biology. In your resubmission, everything is in good order and fine for publication. Below I may suggest another way of wording or add phrases/sentences to your words, or changed the order of a paragraph (discussion). This does not mean that you must take my suggestion/words or use these. They are only suggestions, for you to think about, and possibly use if you think these improve the dialogue. My words are in blue print.

Marianne Jurkowitz

P.S. The blue print of my revisions don't show up here in this window , so I have attached a file below with a word document of my review.

Title and Abstract are much improved and quite good.

Line 35: Does the term “upregulated” mean the same thing as “increased amount of”?

Introduction is also improved.

However in the manuscript, you have shown that lysoplasmalogenase is a highly important enzyme, and the readers may not know the reaction catalyzed by the enzyme, or its history. I think you need to expand on this section and add references suggested below at Line 64.

60 The biosynthesis of plasmalogens begins in the luminal side of the peroxisomal
61 membrane and the intermediate products are further modified into plasmalogens in the
endoplasmic reticulum (ER)⁴

62 . Plasmalogens can be hydrolyzed by phospholipase A2 (PLA2),

63 releasing a fatty acid at the sn-2 position and lysoplasmalogen. The lysoplasmalogen which can be
further metabolized catabolized by phospholipase C (PLC), phospholipase D (PLD), and
lysoplasmalogenase 4, 6, and 8 (please add ref. 6; Why do you have reference 8 here? Please check)

Moreover, lysoplasmalogen may be reacylated at sn-2 by a transacylase enzyme, thus reforming
plasmalogen (Ref. Kramer R. M. and Deykin D. (1983) Arachidonoyl transacylase in human platelets.
Coenzyme A-independent transfer of arachidonate from phosphatidylcholine to
lysoplasmenylethanolamine. J. Biol. Chem. 258, 13806-13811).

(Note to you: This transacylase reaction may be the mechanism by which lysoplasmalogen level is related to the plasmalogen levels in your studies. If lysoplasmalogen levels are low, there are less molecules for reacylation to form plasmalogen. If lysoplasmalogen levels are high, more molecules for reacylation to plasmalogens.)

64 . Lysoplasmalogenases are may be key enzymes that regulate lysoplasmalogen catabolism and lysoplasmalogen and plasmalogen homeostasis (Wu et al., 2011). Lysoplasmalogenase was first identified and characterized in rat liver microsomes as the enzyme that catalyzes the hydrolysis of the vinyl ether bond of choline lysoplasmalogen (Warner H. R. and Lands W. E., (1961) J. Biol. Chem. 236, 2404-2409) or ethanolamine lysoplasmalogen (Gunawan J. and Debuch H. (1981) Hoppe Seylers Z. Physiol. Chem. 362, 445-452) forming fatty aldehyde and glycerophosphocholine or glycerophosphoethanolamine, respectively. The enzyme was later found in rat brain microsomes (Gunawan J. and Debuch H., (1985) J. Neurochem. 44, 370-375) and in small intestinal epithelial cell microsomes (Jurkowitz M. et al., (1999) Biochim et Biophys. Acta 1437, 142-156. The liver enzyme was purified (Jurkowitz-Alexander et al., 1989) and further purified and the gene identified as TMEM86B (Wu et al., 2011).

65, 66, 67

TMEM86B has been characterized as . TMEM86A is a close homolog of the TMEM86B lysoplasmalogenase (6),

68 however, its potential lysoplasmalogenase activity and function in adipose tissue have not been

69 investigated.

Line 76 holoenzyme from heart tissue, and found that

Line 79high-fat feeding upregulated TMEM86A..... Do you think “upregulated” is better than “increased”? I’m not as familiar with the term “upregulated” when you are measuring the amount of protein with a western.

Line 81-83 We first determined provided evidence that TMEM86A is a bona fide lysoplasmalogenase by comprehensive

Line 85-86 We discovered that adipocyte inactivation of adipocyte TMEM86A by gene KO increases oxidative...

Line 88-89... cAMP in adipocytes. cAMP activates Protein Kinase A which subsequently catalyzes phosphorylation of key mitochondrial enzymes responsible for enhanced oxidative phosphorylation and oxidation of nutrient molecules.

Line 89 Importantly, treating mice with lysoplasmalogen, the

Results Section.

Line 112 TMEM86A regulates adipocyte lysoplasmalogen metabolism

A more direct title might be: Evidence that TMEM86A functions as lysoplasmalogenase in adipocytes tissues and cells.(just a suggestion).

Line 114 please add the following reference here in addition to the one reference cited.

Jurkowitz M. S., Azad A. K., Monsma P. C., Keiser T. L., Kanyo J., Lam T.T., Bell C. E., and Schlesinger L. S. (2022) Mycobacterium tuberculosis encode a YhhN family membrane protein with lysoplasmalogenase activity that protects against toxic host lysolipids J. Biol. Chem. 298 (5)

•

Line 138

In your work, you are measuring lysoplasmalogenase activity by the disappearance of lysoplasmalogen. However, lysoplasmalogen could disappear following hydrolysis by PLC, or PLD. However you do have the mutant data and the possible transmembrane structure that supports the hypothesis that TMEM86A is a lysoplasmalogenase and similar to TMEM86B.

My colleagues Vilhelm A. Woltz, Charles E. Bell, and I cloned and expressed the human TMEM86A gene as a C-terminal-GFP-His8-fusion protein into E. coli. Membranes were solubilized, and the protein was purified. It had abundant lysoplasmalogenase activity with both ethanolamine and choline lysoplasmalogen substrates, and its physical and chemical properties including Km and Vm values, and pH profile were very similar to TMEM86B. We have not yet published this work, but I am now writing this research up as a short communication. If you want to use this information as “personal communication” or as “manuscript in preparation” it would be fine with me, but not necessary.

After Line 189

It might be good to summarize- for example,

“These KO studies provide strong evidence that TMEM86A hydrolyzes lysoplasmalogen, thus controlling levels of the lysolipid. Importantly these studies also show that TMEM86A controls levels of plasmalogen, supporting similar findings in HEK cells overexpressing TMEM86B (Wu et al. 2011).”

Lines 263 and 264.

...as major lysoplasmalogens upregulated by TMEM86A KO and down regulated by TMEM86A overexpression

Please check with other scientists about use of terms “up- and down regulated” versus “increased” and “decreased”..

Line 302 For clarity, add word “increased” before the word “mitochondria”

. Line 320 Discussion [SEP]:

. Please state your hypothesis

. The discussion should be more organized.

. My changes in the discussion are just suggested changes. They are not required changes.

. It might be good to start with short overview and hypothesis: We observed that high fat diet causes increase in RNA transcript and protein expression of TMEM86A, a putative lysoplasmalogenase, in adipose tissue and cells. We hypothesized that lysoplasmalgen/lysoplasmalogenase may be involved in lysoplasmalogen metabolism in adipose tissue/cells, and this may relate to obesity and metabolic imbalance and disease.

Our major findings include:

1. TMEM86A functions as a lysoplasmalogenase, and controls the level of lysoplasmalogen in adipose cells and tissue.

. 321 In this research we studied (characterized) the effects of deletion of the TMEM86A gene in a knockout mouse model and the effects of overexpression of TMEM86A gene in adipocytes. [SEP]

. 322 Our methods included analyses of phospholipid composition by untargeted lipidomics.

. Our untargeted global phospholipid These analyses indicated that TMEM86A [SEP]

. 330 overexpression in adipocytes significantly downregulated decreased the amounts of cellular lysoplasmalogenes.

. 331 Moreover in the TMEM86A knockout mouse the levels of lysoplasmalogen in adipose tissue and cells were elevated. These studies demonstrated the impact of TMEM86A on lysoplasmalogen levels and were evidence that TMEM86A functions as a lysoplasmalogenase.

. Furthermore, experiments with [SEP]

. 332 TMEM86A mutants suggested that the highly conserved aspartate residues (D82, D190) in [SEP]

. 333 YhhN proteins (6) are important in lysoplasmalogenase activity of TMEM86A. This is further evidence that TMEM86A is a lysoplasmalogenase.

. and demonstrated the [SEP]

. 323 lysoplasmalogenase activity of TMEM86A in adipocytes. Transcript levels of TMEM86B, the [SEP]

. 324 first discovered enzyme with lysoplasmalogenase activity⁶, were considerably lower than those [SEP]

. 325 of TMEM86A in adipose tissue, suggesting that TMEM86A is a major lysoplasmalogenase in [SEP]

. 326 adipose tissue. TMEM86A expression was also higher in WAT than in BAT. The finding that

transcript and its levels of TMEM86A, but not TMEM86B, [SEP]

. 327 were further induced greatly increased in WAT by HFD feeding strongly suggestings the potential involvement of [SEP]

. 328 TMEM86A in adipose tissue remodeling in response to nutritional stimuli. [SEP]

. 329 lines 329 -333 have been moved up to new location [[Our untargeted global phospholipid analyses indicated that TMEM86A [SEP]

. 330 overexpression in adipocytes significantly downregulated decreased the amounts of lysoplasmalogenes.

. 331 Moreover in the TMEM86A knockout mouse the levels of lysoplasmalogen were elevated. These studies demonstrated the impact of TMEM86A on lysoplasmalogen levels and provide evidence that TMEM86A is a lysoplasmalogenase...

. Furthermore, experiments with [SEP]

. 332 TMEM86A mutants suggested that the highly conserved aspartate residues (D82, D190) in [SEP]

. 333 YhhN proteins (6) are important in lysoplasmalogenase activity of TMEM86A. This is further evidence that TMEM86A is a lysoplasmalogenase.]]

. ... The lipidomics [SEP] analyses in the knockout model and in the overexpressing adipocytes showed that TMEM86A lysoplasmalogenase appears to control the levels of plasmalogen, as well as lysoplasmalogen. Thus plasmalogen levels in adipose cells and tissue correlated with levels of lysoplasmalogen. Over expression of TMEM86B in HEK cells also caused a decrease in plasmalogen in these cells (6). A possible connection between lysoplasmalogen and plasmalogen levels may be related to the transacylation reaction whereby an acyl group is transferred from 1-alkyl-sn-2-acyl-glycerophosphocholine to lysoplasmalogen forming plasmalogen. The enzyme is a transacylase (Kramer R.M. and Deykin D.(1983). When lysoplasmalogen levels are low, there are fewer molecules for reacylation to form plasmalogen. (If lysoplasmalogen levels are high, more molecules are available for reacylation to plasmalogen.)

.

. The lipidomics [SEP]

. 334 analysis also indicated that TMEM86A overexpression upregulated plasmalogen PC and [SEP]

. 335 plasmalogen PE levels. We speculate that the reduced levels of lysoplasmalogen and [SEP]

. 336 plasmalogen indirectly affect the levels of other lipid species involved in plasmalogen [SEP]

. 337 metabolism. It is also possible that TMEM86A has other lipid metabolizing activities beyond [SEP]

. 338 actions as a lysoplasmalogenase. Further analysis of enzyme activity and structural [SEP]

. 339 demonstration of interactions between lipid substrates and TMEM86A protein would be [SEP]

. 340 informative for comprehensive understanding of its physiological function. [SEP]

. 2. Lysoplasmalogen activates the Adenylate Cyclase-cyclicAMP-PKA axis in adipocytes.

. 341 In this study, we demonstrated that an increase in lysoplasmalogen levels by [SEP]

. 342 TMEM86A deficiency facilitated PKA signaling.

. We found that LPE P-18:0 inhibited the phosphodiesterase, PDE3B, [SEP]

. 348 the major enzyme involved in cyclic AMP catabolism. The increase in cyclic-AMP activated c-AMP-dependent-PKA activity, which in turn catalyzed phosphorylation of mitochondrial enzymes and increased mitochondrial oxidative-phosphorylation, and the oxidative catabolism of nutrient molecules. Previous studies indicated that LPC P- [SEP]

. 343 directly increases PKA activity, which was shown by assay with purified PKA proteins from heart muscle in cell- [SEP]

. 344 free system.16. In addition their study found that LPE P- or other lysophospholipid classes do not activate PKA, [SEP]

. 345 indicating the specificity of PKA for lysophospholipids having choline polar head groups. [SEP]

. 346 However, the effects of lysoplasmalogen on PKA-dependent lipid metabolism in the context [SEP]

. 347 of cellular signaling were not determined. Here we found that LPE P-18:0 inhibited PDE3B

. 348 activity and this led to increased cAMP levels in adipocytes. Further investigation is required to understand

. 349 the molecular mechanisms of LPE P-18:0-induced inhibition of PDE3B activity.

. 350 Our current study demonstrated that lysoplasmalogenase deficiency increased

. 351 plasmalogen content, as well as lysoplasmalogen. Plasmalogen metabolism generates diverse

. 352 lipid mediators; thus, several studies have investigated the correlation between plasmalogen

. 353 levels and metabolic disease states. For example, a previous study demonstrated that PC O-

. 354 16:0/18:1 and PC O-16:0/18:2 are inversely associated with waist circumference²³. In

. 355 particular, PC O-16:0/18:1 had a positive relationship with c-peptide and adiponectin, which

. 356 suggested its association with improved insulin signaling and glucose metabolism²³. (Was LPC (O-16:0) measured in that study? It is possible that LPC (O-16:0) was also elevated, and was the reason for the lean body state.) An in vivo

. 357 study performed by Park et al. reported that adipocyte-specific deletion of Pex16, a crucial

. 358 factor for peroxisome biogenesis that is responsible for plasmalogen synthesis, led to the

. 359 prevention of mitochondrial fission, impaired thermogenesis, and reduced peroxisomes in

. 360 adipocytes⁷. Supplementing Pex16 KO mice with plasmalogens rescued mitochondrial

. 361 function and thermogenesis⁷. In contrast, independent studies have demonstrated that high

. 362 plasmalogen and arachidonic acid content in the adipose tissue of obese individuals increases

. 363 vulnerability to inflammation⁵. These seemingly contradictory results may be due to the

. 364 complex nature of plasmalogen structure and metabolism, which are uniquely attributed to

. 365 tissue types and developmental stages.

. 366 In the current study, we found that deletion of TMEM86A in adipocytes protects mice

. 367 from obesity and insulin resistance in vivo. This was accompanied by improved glucose

. 368 tolerance and insulin sensitivity and reduced pro-inflammatory macrophage polarization.

. 369 Several mouse model studies have reported that increase in lipid catabolism and mitochondrial

. 370 activity in the adipose tissue is beneficial in protection against HFD-induced insulin resistance

. 371 and inflammation²⁴. Thus, it is reasonable to consider that lysoplasmalogen-induced PKA

. 372 signaling in the adipose tissue could be therapeutically targeted for overcoming obesity-

. 373 induced metabolic dysfunction^{24, 25}.

. 374 In summary, our data suggest the important role of TMEM86A-mediated

. 375 lysoplasmalogen metabolism in adipose tissue function and indicate a potential therapeutic

. 376 strategy for obesity-related metabolic diseases.

End of review Marianne S. Jurkowitz

Reviewer #2 (Remarks to the Author):

All the comments I have made were addressed by the Authors.

Reviewer #3 (Remarks to the Author):

The authors have adequately addressed all of my critiques and I have no further questions.

We thank the Editors and Reviewers for their valuable comments and the opportunity to improve our manuscript. We revised the manuscript to incorporate Reviewer 1's suggestions and editorial requests, as indicated in point-by-point responses.

We provided a two-sentence summary of the current work as requested in the checklist. "Dysregulation of plasmalogen metabolism in adipose tissue is associated with the development of metabolic diseases. The current study characterized the role of adipocyte TMEM86A as a lysoplasmalogenase in adipose tissues, and highlighted the therapeutic potential of targeting this pathway in obesity-related metabolic diseases."

In the data availability statement, we included accession codes and web links for publicly available datasets. The lipidomics data generated in this study had been deposited in a public database, MetaboLights (accession # MTBLS4703), and the data will be publicly released after "in curation" period. Currently, we cannot be sure how long this "in curation" process might take. Therefore, we provided ID and password in the Reporting Summary and the data availability statement to enable any access to the data MTBLS4703 (ID: yunyochl@snu.ac.kr, PW: NCOMMS-22-00143)

REVIEWERS' COMMENTS

Reviewer #1 (Remarks to the Author):

20220517

Review of resubmission.

Dear Authors,

You have addressed all of my concerns or changes I suggested in the first submission. This research work is highly important contribution to biology. In your resubmission, everything is in good order and fine for publication. Below I may suggest another way of wording or add phrases/sentences to your words, or changed the order of a paragraph (discussion). This does not mean that you must take my suggestion/words or use these. They are only suggestions, for you to think about, and possibly use if you think these improve the dialogue. My words are in blue print.

Marianne Jurkowitz

Responses: We thank the reviewer Dr. Marianne Jurkowitz for the review and helpful comments. We revised the manuscript to incorporate the reviewer's suggestions, as indicated below.

P.S. The blue print of my revisions don't show up here in this window, so I have attached a file below with a word document of my review.

Title and Abstract are much improved and quite good.

Line 35: Does the term "upregulated" mean the same thing as "increased amount of"?

Response: We changed "upregulated" into "increased"

Introduction is also improved.

However in the manuscript, you have shown that lysoplasmalogenase is a highly important enzyme, and the readers may not know the reaction catalyzed by the enzyme, or its history. I think you need to expand on this section and add references suggested below at Line 64.

Response: We revised and expanded this section according to the comments.

60 The biosynthesis of plasmalogens begins in the luminal side of the peroxisomal
61 membrane and the intermediate products are further modified into plasmalogens in the
endoplasmic reticulum (ER)⁴

62 Plasmalogens can be hydrolyzed by phospholipase A2 (PLA2),
63 releasing a fatty acid at the sn-2 position and lysoplasmalogen. The lysoplasmalogen
which can be further metabolized catabolized by phospholipase C (PLC), phospholipase D
(PLD), and lysoplasmalogenase 4, 6, and 8 (please add ref. 6; Why do you have reference 8
here? Please check)

Response: We replaced Ref. 8 with more relevant reference Ref. 6 (Ref. 8 was removed).

Moreover, lysoplasmalogen may be reacylated at sn-2 by a transacylase enzyme, thus
reforming plasmalogen (Ref. Kramer R. M. and Deykin D. (1983) Arachidonoyl transacylase
in human platelets. Coenzyme A-independent transfer of arachidonate from
phosphatidylcholine to lysoplasmenylethanolamine. J. Biol. Chem. 258, 13806-13811).

Response: We included the citation.

(Note to you: This transacylase reaction may be the mechanism by which lysoplasmalogen
level is related to the plasmalogen levels in your studies. If lysoplasmalogen levels are low,
there are less molecules for reacylation to form plasmalogen. If lysoplasmalogen levels are
high, more molecules for reacylation to plasmalogens.)

64 Lysoplasmalogenases are may be key enzymes that regulate lysoplasmalogen
catabolism and lysoplasmalogen and plasmalogen homeostasis (Wu et al., 2011).
Lysoplasmalogenase was first identified and characterized in rat liver microsomes as the
enzyme that catalyzes the hydrolysis of the vinyl ether bond of choline lysoplasmalogen
(Warner H. R. and Lands W. E., (1961) J. Biol. Chem. 236, 2404-2409) or ethanolamine
lysoplasmalogen (Gunawan J. and Debuch H. (1981) Hoppe Seylers Z. Physiol. Chem. 362,
445-452) forming fatty aldehyde and glycerophosphocholine or
glycerophosphoethanolamine, respectively. The enzyme was later found in rat brain
microsomes (Gunawan J. and Debuch H., (1985) J. Neurochem. 44, 370-375) and in small
intestinal epithelial cell microsomes (Jurkowitz M. et al., (1999) Biochim et Biophys. Acta
1437, 142-156. The liver enzyme was purified (Jurkowitz-Alexander et al., 1989) and further
purified and the gene identified as TMEM86B (Wu et al., 2011).

Response: All edits and the new sentences were integrated into the manuscript as the
reviewer suggested.

65, 66, 67

TMEM86B has been characterized as . TMEM86A is a close homolog of the TMEM86B
lysoplasmalogenase (6),

68 however, its potential lysoplasmalogenase activity and function in adipose tissue have not
been

69 investigated.

Line 76 holoenzyme from heart tissue, and found that

Line 79high-fat feeding upregulated TMEM86A..... Do you think "upregulated" is better
than "increased"? I'm not as familiar with the term "upregulated" when you are measuring the
amount of protein with a western.

Line 81-83 We first determined provided evidence that TMEM86A is a bona fide
lysoplasmalogenase by comprehensive

Line 85-86 We discovered that adipocyte inactivation of adipocyte TMEM86A by gene KO
increases oxidative...

Line 88-89... cAMP in adipocytes. cAMP activates Protein Kinase A which subsequently

catalyzes phosphorylation of key mitochondrial enzymes responsible for enhanced oxidative phosphorylation and oxidation of nutrient molecules.

Line 89 Importantly, treating mice with lysoplasmalogen, the

Results Section.

Line 112 TMEM86A regulates adipocyte lysoplasmalogen metabolism

A more direct title might be: Evidence that TMEM86A functions as lysoplasmalogenase in adipocytes tissues and cells.(just a suggestion).

Response: We changed the subtitle as suggested by the reviewer.

Line 114 please add the following reference here in addition to the one reference cited.

Jurkowitz M. S., Azad A. K., Monsma P. C., Keiser T. L., Kanyo J., Lam T.T., Bell C. E., and Schlesinger L. S. (2022) Mycobacterium tuberculosis encode a YhhN family membrane protein with lysoplasmalogenase activity that protects against toxic host lysolipids J. Biol. Chem. 298 (5)

Response: We included the citation.

Line 138

In your work, you are measuring lysoplasmalogenase activity by the disappearance of lysoplasmalogen. However, lysoplasmalogen could disappear following hydrolysis by PLC, or PLD. However you do have the mutant data and the possible transmembrane structure that supports the hypothesis that TMEM86A is a lysoplasmalogenase and similar to TMEM86B.

My colleagues Vilhelm A. Woltz, Charles E. Bell, and I cloned and expressed the human TMEM86A gene as a C-terminal-GFP-His8-fusion protein into E. coli. Membranes were solubilized, and the protein was purified. It had abundant lysoplasmalogenase activity with both ethanolamine and choline lysoplasmalogen substrates, and its physical and chemical properties including Km and Vm values, and pH profile were very similar to TMEM86B. We have not yet published this work, but I am now writing this research up as a short communication. If you want to use this information as “personal communication” or as “manuscript in preparation” it would be fine with me, but not necessary.

Response: In the Discussion we cite as ‘personal communication’ Dr. Jurkowitz’s unpublished work concerning the physical and chemical properties of purified recombinant TMEM86A protein.

After Line 189

It might be good to summarize- for example,

“These KO studies provide strong evidence that TMEM86A hydrolyzes lysoplasmalogen, thus controlling levels of the lysolipid. Importantly these studies also show that TMEM86A controls levels of plasmalogen, supporting similar findings in HEK cells overexpressing TMEM86B (Wu et al. 2011).”

Response: We include the sentence to summarize the results.

Lines 263 and 264.

...as major lysoplasmalogens upregulated by TMEM86A KO and down regulated by TMEM86A overexpression

Please check with other scientists about use of terms “up- and down regulated” versus “increased” and “decreased”.

Response: We changed “upregulated” into “increased”.

Line 302 For clarity, add word “increased” before the word “mitochondria”

Response: We added the word “increased”.

- . Line 320 Discussion [SEP]
- . Please state your hypothesis
- . The discussion should be more organized.
- . My changes in the discussion are just suggested changes. They are not required changes.
- . It might be good to start with short overview and hypothesis: We observed that high fat diet causes increase in RNA transcript and protein expression of TMEM86A, a putative lysoplasmalogenase, in adipose tissue and cells. We hypothesized that lysoplasmalgen/lysoplasmalogenase may be involved in lysoplasmalogen metabolism in adipose tissue/cells, and this may relate to obesity and metabolic imbalance and disease.

Response: We revised the “Discussion” as suggested by the reviewer. We started the “Discussion” with a short overview and hypothesis.

- ⇒ “Dysregulation of plasmalogen metabolism in adipose tissue has been implicated in the development of obesity-related metabolic diseases, yet little is known about the mechanisms regulating plasmalogen homeostasis in adipocytes or their impact on systemic metabolism. We observe that HFD increased RNA transcript and protein expression of TMEM86A, a putative lysoplasmalogenase in adipose tissue. We hypothesized that lysoplasmalogenase might be involved in plasmalogen metabolism in adipose tissue, and this may relate to obesity and metabolic imbalance and dysfunction. In this research, we.....”

Our major findings include:

1. TMEM86A functions as a lysoplasmalogenase, and controls the level of lysoplasmalogen in adipose cells and tissue.

. 321 In this research we studied (characterized) the effects of deletion of the TMEM86A gene in a knockout mouse model and the effects of overexpression of TMEM86A gene in adipocytes. [SEP]

. 322 Our methods included analyses of phospholipid composition by untargeted lipidomics.

. Our untargeted global phospholipid These analyses indicated that TMEM86A [SEP]

. 330 overexpression in adipocytes significantly downregulated decreased the amounts of cellular lysoplasmalogenes.

. 331 Moreover in the TMEM86A knockout mouse the levels of lysoplasmalogen in adipose tissue and cells were elevated. These studies demonstrated the impact of TMEM86A on lysoplasmalogen levels and were evidence that TMEM86A functions as a lysoplasmalogenase.

. Furthermore, experiments with [SEP]

. 332 TMEM86A mutants suggested that the highly conserved aspartate residues (D82, D190) in [SEP]

. 333 YhhN proteins (6) are important in lysoplasmalogenase activity of TMEM86A. This is further evidence that TMEM86A is a lysoplasmalogenase.

. and demonstrated the [SEP]

. 323 lysoplasmalogenase activity of TMEM86A in adipocytes. Transcript levels of TMEM86B, the [SEP]

. 324 first discovered enzyme with lysoplasmalogenase activity6, were considerably lower than those [SEP]

. 325 of TMEM86A in adipose tissue, suggesting that TMEM86A is a major lysoplasmalogenase in [SEP]

. 326 adipose tissue. TMEM86A expression was also higher in WAT than in BAT. The finding that transcript and its levels of TMEM86A, but not TMEM86B, [SEP]

. 327 were further induced greatly increased in WAT by HFD feeding strongly suggesting the potential involvement of [SEP]

- . 328 TMEM86A in adipose tissue remodeling in response to nutritional stimuli. [SEP]
- . 329 lines 329 -333 have been moved up to new location [[Our untargeted global phospholipid analyses indicated that TMEM86A [SEP]
- . 330 overexpression in adipocytes significantly downregulated decreased the amounts of lysoplasmalogens.
- . 331 Moreover in the TMEM86A knockout mouse the levels of lysoplasmalogen were elevated. These studies demonstrated the impact of TMEM86A on lysoplasmalogen levels and provide evidence that TMEM86A is a lysoplasmalogenase...
- . Furthermore, experiments with [SEP]
- . 332 TMEM86A mutants suggested that the highly conserved aspartate residues (D82, D190) in [SEP]
- . 333 YhhN proteins (6) are important in lysoplasmalogenase activity of TMEM86A. This is further evidence that TMEM86A is a lysoplasmalogenase.]]
- The lipidomics [SEP]analyses in the knockout model and in the overexpressing adipocytes showed that TMEM86A lysoplasmalogenase appears to control the levels of plasmalogen, as well as lysoplasmalogen. Thus plasmalogen levels in adipose cells and tissue correlated with levels of lysoplasmalogen.

Over expression of TMEM86B in HEK cells also caused a decrease in plasmalogens in these cells (6). A possible connection between lysoplasmalogen and plasmalogen levels may be related to the transacylation reaction whereby an acyl group is transferred from 1-alkyl-sn-2-acyl-glycerophosphocholine to lysoplasmalogen forming plasmalogen. The enzyme is a transacylase (Kramer R.M. and Deykin D.(1983). When lysoplasmalogen levels are low, there are fewer molecules for reacylation to form plasmalogen. (If lysoplasmalogen levels are high, more molecules are available for reacylation to plasmalogens.)

Response: we incorporated the reviewer's insightful comments and revised the paragraph in the Discussion, as indicated below.

- ⇒ The lipidomics analyses in the KO mouse model and in the overexpressing adipocytes showed that TMEM86A lysoplasmalogenase appears to control the levels of both lysoplasmalogens and plasmalogens. Consistently, previous work reported that overexpression of TMEM86B decreases both lysoplasmalogens and plasmalogens in HEK cells⁶. While the mechanistic connection between plasmalogen and lysoplasmalogen levels is not known, it is possible that loss of lysoplasmalogens reduces the abundance of acyl groups available for transacylation of lysoplasmalogen to plasmalogens.

-
- . The lipidomics [SEP]
- . 334 analysis also indicated that TMEM86A overexpression upregulated plasmalogen PC and [SEP]
- . 335 plasmalogen PE levels. We speculate that the reduced levels of lysoplasmalogens and [SEP]
- . 336 plasmalogens indirectly affect the levels of other lipid species involved in plasmalogen [SEP]
- . 337 metabolism. It is also possible that TMEM86A has other lipid metabolizing activities beyond [SEP]
- . 338 actions as a lysoplasmalogenase. Further analysis of enzyme activity and structural [SEP]
- . 339 demonstration of interactions between lipid substrates and TMEM86A protein would be [SEP]
- . 340 informative for comprehensive understanding of its physiological function. [SEP]
- . 2. Lysoplasmalogen activates the Adenylate Cyclase-cyclicAMP-PKA axis in adipocytes.
- . 341 In this study, we demonstrated that an increase in lysoplasmalogen levels by [SEP]
- . 342 TMEM86A deficiency facilitated PKA signaling.
- . We found that LPE P-18:0 inhibited the phosphodiesterase, PDE3B, [SEP]
- . 348 the major enzyme involved in cyclic AMP catabolism. The increase in cyclic-AMP

activated c-AMP-dependent-PKA activity, which in turn catalyzed phosphorylation of mitochondrial enzymes and increased mitochondrial oxidative-phosphorylation, and the oxidative catabolism of nutrient molecules.

Response: The above sentence suggested by the reviewer includes misinterpretation of the results (e.g. “phosphorylation of mitochondrial enzymes” was not examined in our current study), so we did not include the sentence. The errors are corrected and the following sentences were added.

“In turn, enhanced cAMP-dependent phosphorylation of key metabolic enzymes and transcription factors, such as HSL and CREB, promoted expression of mitochondrial electron transport enzymes and expanded oxidative catabolism of adipocytes. Mechanistically, we found that LPE P-18:0 inhibited the phosphodiesterase, PDE3B, the major enzyme involved in cAMP catabolism.”

Previous studies indicated that LPC P-

. 343 directly increases PKA activity, which was shown by assay with purified PKA proteins from heart muscle in cell-

. 344 free system.¹⁶ In addition their study found that LPE P- or other lysophospholipid classes do not activate PKA,

. 345 indicating the specificity of PKA for lysophospholipids having choline polar head groups.

. 346 However, the effects of lysoplasmalogen on PKA-dependent lipid metabolism in the context

. 347 of cellular signaling were not determined. Here we found that LPE P-18:0 inhibited PDE3B

. 348 activity and this led to increased cAMP levels in adipocytes. Further investigation is required to understand

. 349 the molecular mechanisms of LPE P-18:0-induced inhibition of PDE3B activity.

. 350 Our current study demonstrated that lysoplasmalogenase deficiency increased

. 351 plasmalogen content, as well as lysoplasmalogen. Plasmalogen metabolism generates diverse

. 352 lipid mediators; thus, several studies have investigated the correlation between plasmalogen

. 353 levels and metabolic disease states. For example, a previous study demonstrated that PC O-

. 354 16:0/18:1 and PC O-16:0/18:2 are inversely associated with waist circumference²³. In

. 355 particular, PC O-16:0/18:1 had a positive relationship with c-peptide and adiponectin, which

. 356 suggested its association with improved insulin signaling and glucose metabolism²³. (Was LPC (O-16:0) measured in that study? It is possible that LPC (O-16:0) was also elevated, and was the reason for the lean body state.)

Response: We thank the reviewer for the insightful comments. We examined LPC-O 16:0, but we did not detect a significant difference in LPC-O 16:0 levels between lean and obese groups.

Response: The following paragraphs are the copied text from our manuscript. There were no further questions or comments from the reviewer below.

An in vivo

. 357 study performed by Park et al. reported that adipocyte-specific deletion of Pex16, a crucial

. 358 factor for peroxisome biogenesis that is responsible for plasmalogen synthesis, led to the

. 359 prevention of mitochondrial fission, impaired thermogenesis, and reduced peroxisomes

in

. 360 adipocytes⁷. Supplementing Pex16 KO mice with plasmalogens rescued mitochondrial
function and thermogenesis⁷. In contrast, independent studies have demonstrated that
high plasmalogen and arachidonic acid content in the adipose tissue of obese individuals
increases vulnerability to inflammation⁵. These seemingly contradictory results may be due to the
complex nature of plasmalogen structure and metabolism, which are uniquely
attributed to different tissue types and developmental stages.
In the current study, we found that deletion of TMEM86A in adipocytes protects mice
from obesity and insulin resistance in vivo. This was accompanied by improved
glucose tolerance and insulin sensitivity and reduced pro-inflammatory macrophage
polarization.
Several mouse model studies have reported that increase in lipid catabolism and
mitochondrial activity in the adipose tissue is beneficial in protection against HFD-induced insulin
resistance and inflammation²⁴. Thus, it is reasonable to consider that lysoplasmalogen-induced
PKA signaling in the adipose tissue could be therapeutically targeted for overcoming
obesity-induced metabolic dysfunction^{24, 25}.
In summary, our data suggest the important role of TMEM86A-mediated
lysoplasmalogen metabolism in adipose tissue function and indicate a potential
therapeutic strategy for obesity-related metabolic diseases.
End of review Marianne S. Jurkowitz

Reviewer #2 (Remarks to the Author):

All the comments I have made were addressed by the Authors.

Response: We thank the Reviewer for the valuable comments and the opportunity to improve our manuscript.

Reviewer #3 (Remarks to the Author):

The authors have adequately addressed all of my critiques and I have no further questions.

Response: We thank the Reviewer for the valuable comments and the opportunity to improve our manuscript.